# The Expressive Capacity of State Space Models: A Formal Language Perspective

**Yash Sarrof, Yana Veitsman, Michael Hahn**
Saarland Informatics Campus
Saarland University, Germany
`{ysarrof, yanav, mhahn}@lst.uni-saarland.de`

## Abstract

Recently, recurrent models based on linear state space models (SSMs) have shown promising performance in language modeling (LM), competititve with transformers. However, there is little understanding of the in-principle abilities of such models, which could provide useful guidance to the search for better LM architectures. We present a comprehensive theoretical study of the capacity of such SSMs as it compares to that of transformers and traditional RNNs. We find that SSMs and transformers have overlapping but distinct strengths. In star-free state tracking, SSMs implement length-generalizing solutions to problems that transformers struggle to represent exactly. They can also model bounded hierarchical structure with optimal memory even without simulating a stack. On the other hand, we identify a design choice in current SSMs that limits their expressive power. We discuss implications for SSM and LM research, and verify results empirically [1] on a recent SSM, Mamba.

## 1 Introduction

Transformers [Vaswani et al., 2017] power most large language models (LLMs) today, as they offer the advantage of parallelized training by avoiding recurrence, compared to the previously dominant recurrent achitectures [RNNs Elman, 1990, Hochreiter and Schmidhuber, 1997]. However, building on a long history of continuous dynamical models [e.g. Kalman, 1960, 1963] and work on faster RNNs [Bradbury et al., 2016, Lei et al., 2018], a recent line of work has developed *state space models* (SSMs) rivaling the performance of transformers [e.g. Gu et al., 2021, Gu and Dao, 2023, Sun et al., 2023, De et al., 2024, Yang et al., 2024, Qin et al., 2024a]. These SSMs are recurrent models, formulated in terms of iterative state updates, while still allowing efficient parallelization.

The impressive empirical performance of such SSMs raises the question of whether they might have capabilities that the transformer architecture might lack in principle. Simultaneously, to understand whether SSMs may plausibly overtake the dominant role of transformers, it is an important question whether SSMs may lack abilities present in transformers. A better understanding of these questions may also point the way to future architectures that unite the strengths of both architectures.

One common approach to understanding the capabilities of computational architectures is through their expressive capacity in simulating automata and modeling language classes; indeed, a sizeable literature has studied transformers [e.g. Pérez et al., 2019, Hahn, 2020, Bhattamishra et al., 2020, Yao et al., 2021a, Liu et al., 2023b,a, Deletang et al., 2022, Strobl et al., 2024, Chiang et al., 2023, Sanford et al., 2024, Peng et al., 2024] and RNNs [e.g. Siegelman and Sontag, 1995, Horne and Hush, 1993, Indyk, 1995, Weiss et al., 2018, Hewitt et al., 2020] through this lens. As the difficulty

---

[1]Code is available at: https://github.com/lacoco-lab/ssm_expressivity

38th Conference on Neural Information Processing Systems (NeurIPS 2024).

of many computational problems is well-understood in terms of such language classes, results about expressive capacity directly yield results about the ability to model specific computational problems.

While a substantial number of results have been obtained for transformers and traditional RNNs, understanding remains largely open for SSMs. In an initial step, Merrill et al. [2024] showed that all problems computable by SSMs are contained in $TC^0$, a circuit complexity class that is known to also cover transformers [Merrill and Sabharwal, 2023, Strobl, 2023]. Under standard conjectures, this suggests that certain types of state tracking are hard for both models. Jelassi et al. [2024] and Bhattamishra et al. [2024] provided evidence of differences between these architectures, showing that transformers outperform SSMs on copying or retrieving from long strings–tasks well within $TC^0$. Zubić et al. [2024] showed that multi-layer SSMs are constrained by their logarithmic space computational capacity, limiting their ability at algorithmic tasks such as multi-digit multiplication.

However, a more fine-grained understanding of the power of SSMs, and how they compare to RNNs and transformers, remains an open question. Our contribution in this paper is to provide rigorous understanding of SSMs' abilities in different classes of languages. We show that transformers and SSMs cover overlapping but distinct fragments of $TC^0$. For instance, SSMs can model bounded hierarchical structure in ways similar to transformers and traditional RNNs, even without embedding a stack-like structure (Theorem 6). For regular languages involving modular counting, such as the PARITY function (Theorem 2), we identify a design choice that makes extant SSMs struggle in ways similar to transformers. In other cases, we show that SSMs resolve a failure case of transformers: they effortlessly model Flip Flop state tracking (Theorem 1). We discuss take-aways for SSM and LLM research in Section 5; among others, our results suggest future LM architectures might need to combine both attention and state spaces.

## 2 Background

### 2.1 State Space Models

**SSM Layers** We define a single layer of a state space model as a map, at input length $T$,

$$\mathbb{R}^{T \times d} \to \mathbb{R}^{T \times d} \qquad\qquad (x_t)_{t=1,\dots,T} \mapsto (z_t)_{t=1,\dots,T}$$

given by the recurrence

$$h_t = A(x_t) \circ h_{t-1} + B(x_t) \qquad\qquad z_t = \phi(h_t, x_t) \qquad\qquad (1)$$

where $\circ$ denotes elementwise product, and, for each $x_t \in \mathbb{R}^d$,

$$h_0 \in \mathbb{R}^d \qquad\qquad B(x_t) \in \mathbb{R}^d \text{ (increment)}$$
$$A(x_t) \in \mathbb{R}^d \text{ (gate)} \qquad\qquad \phi : \mathbb{R}^{2d} \to \mathbb{R}^d \text{ (transform)}$$

We allow $A, B$ to be arbitrary smooth maps. The map $\phi(h_t, x_t)$ includes a cascade of channel-mixing transformations and normalization, which we abstract as follows:

$$\phi(h_t, x_t) = \text{Mix}_1(\text{Norm}(\text{Mix}_2(h_t, x_t)), x_t) \qquad\qquad (2)$$

where $\text{Mix}_j(\cdot)$ can contain linear or (Swi)GLU components [e.g. Qin et al., 2024a, Gu and Dao, 2023]. We will take Norm to implement RMSNorm Zhang and Sennrich [2019]; LayerNorm Ba et al. [2016] can be covered by absorbing centering into $\text{Mix}_2$.

**A Full SSM** Real-world SSMs typically stack several layers of the form (1–2). Where needed, we use superscripts to indicate the layers in an SSM: $h_t^{(1)}, \dots, h_t^{(L)}$, where $L$ is the number of layers. We consider input words $\mathbf{w} = w_{1\dots|w|}$ over a discrete alphabet $\Sigma$, and assume an encoding in terms of token embeddings $e(\sigma) \in \mathbb{R}^d$, for $\sigma \in \Sigma$. We will also write $e_\sigma$ for $e(\sigma)$. These feed into the lowest layer as $x_t^{(1)} := e(w_t)$. The outputs of each layer feed into the next layer, as $x_t^{(l+1)} = z_t^{(l)}$. The transformations in (1) are specific to each layer: $A^{(1)}, \dots, A^{(L)}$ and similarly for $B, \phi$. To keep notation simple, we will only show the superscripts where necessary for disambiguation. The activations $z_t^{(L)}$ at the highest layer are read out by some neural network $\rho$ into vectors $q_t \in \mathbb{R}^{d_{pred}}$ describing classification or next-token predictions. We again take $\rho$ to be an arbitrary function; importantly, all our constructions will allow $\rho$ to operate correctly even at finite precision.

**Implementation Choices**  In Mamba, (1) directly maps onto Eqs. (2a) and (2b) in Gu and Dao [2023]. The notation of Gu and Dao [2023] use a matrix multiplication $\overline{A}h_{t-1}$ instead of elementwise multiplication $A(x_t) \circ h_{t-1}$ in (1), but importantly, Mamba's $\overline{A}$ is diagonal, so we can take $A(x_t)_i = \overline{A}_{ii}$. Some SSMs assume nondiagonal $A(x_t)$, but typically this matrix is diagonalizable [e.g. Gu et al., 2021, Sun et al., 2023], so that the SSM is still equivalent to one of the form (1). We discuss how other SSMs instantiate (1) in Appendix A. Some models assume complex-valued activations (Appendix A); our results largely do not depend on this distinction, but take it into account where needed (Theorem 13). Some SSMs [e.g. Gu and Dao, 2023] use different numbers of channels in $x_t$ and $h_t$ using state expansion; as this does not affect expressive capacity, we will simply assume a constant dimensionality $d$. Local convolutions [e.g. Fu et al., 2023] can be simulated with an SSM layer and do not increase expressive capacity (Remark 19).

We will find that two design choices have nontrivial impact on expressive capacity: The first one is time invariance: we call an SSM TIME-INVARIANT if $A(x_t)$ does not depend on $x_t$. Some SSMs, such as S4 [Gu et al., 2021] and Retnet [Sun et al., 2023] are time-invariant; Mamba [Gu and Dao, 2023], Griffin [De et al., 2024], GLA [Yang et al., 2024], HGRN [Qin et al., 2024b,a], QRNN/SRU Bradbury et al. [2016], Lei et al. [2018] are not (Appendix A). The second one is the sign of the entries of $A(x_t)$: Across all non-time-invariant SSMs surveyed, we find that the gate is always non-negative (Appendix A): $A(x_t) \geq 0$ (NONNEGATIVE) due to exponential or sigmoid parameterizations of the gate – this choice turns out to limit expressive capacity (Theorem 2).

**Role of Parameterization**  While the abstract form (1–2) is common across the SSM literature, differences in parameterization may have substantial effect on efficiency and training stability. In particular, the parameterization of $A(x_t)$ has been the subject of substantial research [e.g. Gu et al., 2020, 2021, Yu et al., 2023, Wang and Li, 2023]. However, studying expressiveness allows us to abstract away from these differences to a remarkable degree: We will allow $A, B, \rho$ to be *arbitrary* functions with the given input-output properties. Our negative results are based on abstract properties of the setup (1–2), which fundamentally bottlenecks SSMs through *elementwise linear* state updates. For our positive results, will use empirical learnability experiments to verify that learnable solutions instantiating them (though not necessarily implementing the same constructions as used in the proofs) do exist in a recent SSM [Mamba, Gu and Dao, 2023].

We contrast SSMs with traditional RNNs such as simple RNNs or LSTMs: for these, the recurrence in Eq. (1) is replaced by $h_t = \psi(h_{t-1}, x_t)$ where $\psi$ could be linear, an MLP [Elman, 1990], or a more complex gated function [Hochreiter and Schmidhuber, 1997].

**Finite Precision Assumption**  While Eq.(1) assumes arbitrary real-valued activations, real-world implementations can only represent numbers with bounded precision. Formally, we adopt the *finite precision* notion used by Weiss et al. [2018] in a study of the expressive power of traditional RNNs: We allow an unbounded number of integer bits, but only $p$ fractional bits, independent of the length of the input. See Appendix E for discussion.

## 2.2 Modeling Formal languages

We study three foundational types of data structures needed for modeling formal languages [Hopcroft et al., 2001]: finite state automata (Theorem 1, 2, 4), counters (Theorem 5), and stacks (Theorem 6). These data structures can be understood in two equivalent forms: One is to track a state sequence over an input, where each input symbol engenders a specific transformation on the state. The other one, more commonly considered in research on expressive capacity, considers *formal languages*—sets of finite strings that are defined by the property that an automaton reaches one of a pre-specified set of "accepting" states after traversing the word. We focus on the latter, enabling easy comparison with existing results on transformers and RNNs.

A **finite-state-automaton** (see Definition 7) represents a general state tracking problem over a finite state space, without imposing further structure on the state space: The automaton keeps track of a single state from a finite state space; when reading a string from left to right, each symbol engenders a specific transformation of the state. At each position, the current state determines which symbols can come next; membership in a formal language is determined by the state reached after reading the full string. Finite-state-automata are equivalent in expressivity to regular expressions, and define the **regular languages** [Kleene, 1951].

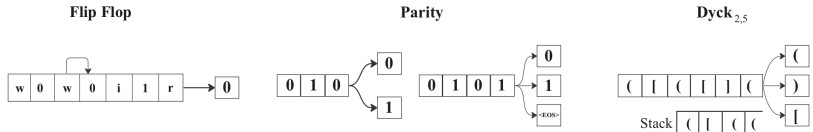

Figure 1: Three key formal languages: prefixes with the sets of possible next characters: Flip Flop (Theorem 1), PARITY (Theorem 2), bounded-depth Dyck (Theorem 6). In Flip Flop, after a `r` (read) instruction, the bit must match what came after the last `w` (write) instruction (here, 0). For PARITY, EOS can only follow when the number of ones in the prefix is even. For bounded-depth Dyck, a closing bracket can only appear if it matches the last unclosed opening bracket (here, ")" matches "(")). Opening brackets can appear as long as the maximum depth (here, 5) hasn't been reached.

Allowing an automaton to keep track of one or more **counters** [Fischer et al., 1968b]—integers that are incremented or decremented at each symbol read—turns the state space infinite, but in a highly structured manner. SSMs can model this datastructure (Theorem 5), as can RNNs and transformers [Weiss et al., 2018, Bhattamishra et al., 2020]. **Stacks**, a first-in-first-out datastructure, enable automata to keep track of hierarchical structure, foundational to natural language [Chomsky, 1957]. We show that SSMs can implement shortcut solutions to *bounded* hierarchical structure even without implementing a stack (Theorem 6) – these are likely to be most useful to natural language given the boundedness of human memory [Miller, 1963, Karlsson, 2007].

### 2.3 Formal Language Prediction and Recognition

We fix a finite alphabet $\Sigma$. Its elements are called *characters* or *symbols*. The set of all finite strings **w** over $\Sigma$ is denoted $\Sigma^*$; such strings are often referred to as *words*. The length of **w** is denoted $|\mathbf{w}|$. A *formal language L* is a subset of $\Sigma^*$. Techically, we assume that the alphabet includes BOS and EOS symbols, which occurs at the beginning and end of each element of $L$ and nowhere else.

We next need to define what it means for an SSM to model a formal language. The notion of *recognition*, where the task is to classify a full string as belonging to the language or not. Formally, for an SSM with $d_{pred} = 1$, we say that it **recognizes** a language $L$ if the output $\rho(z_{|\mathbf{w}|}^{(L)})$ equals—when the SSM is run on $\mathbf{w} \in \Sigma^*$—1 if $\mathbf{w} \in L$ and 0 else.

However, such a classification task is arguably not always matched to dominant use cases in predictive sequence modeling, where the task is to predict the next token at each step. Thus, we also cast formal languages into a language modeling and sequence prediction framework. We adopt the task of Bhattamishra et al. [2020], where the model is asked to output at each step in a sequence the set of possible next symbols. Let $\text{Prefix}(L) := \{w : w \in \Sigma^*, w\Sigma^* \cap L \neq \emptyset\}$ the set of valid prefixes of *L*. We then say that a model **predictively models** a language *L* if (Figure 1), given a valid prefix $w \in \text{Prefix}(L)$, it outputs the finite set

$$\{\sigma \in \Sigma : w\sigma\Sigma^* \cap L \neq \emptyset\} \tag{3}$$

We think of each such set as an atomic label; the set of possible labels is the power set of the finite alphabet $\Sigma$ (here, $d_{pred} = 2^{|\Sigma|}$). Importantly, in both recognition and predictive modeling, we test the SSMs' ability across arbitrary input lengths, i.e. the choice of input length does not affect the inherent capability to recognize or predictively model the language. Predictive modeling can be easily converted into recognition by checking whether any symbol in the sequence is not in the predictive set at the preceding position; this can be done by adding 1 SSM layer. Conversely, if we can show that SSMs cannot recognize a language, this proves they also cannot perform predictive modeling for it, as they then cannot correctly predict where EOS can appear. To get the strongest results, we thus prove positive results for *predictive modeling*, and negative results for *recognition*.

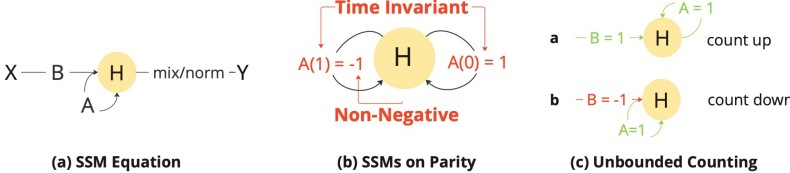

**(a) SSM Equation**   **(b) SSMs on Parity**   **(c) Unbounded Counting**

Figure 2: (a) Visualizing the SSM equations 1, 2: The hidden state $H$ is updated by a combination of its previous values, transformed by matrix $A$, and the input $X$, modulated by matrix $B$. The updated hidden state and input are then processed through a $Mix(.)$ layer, which can incorporate components like (Swi)GLU or Linear layers, with an optional RMSNorm for normalization. (b) An intuitive construction for recognizing PARITY with SSMs is achieved by setting $B = 0$ and $A = -1$ when the input is 1, and $A = 1$ otherwise. However, this construction violates both NONNEGATIVE and TIME-INVARIANT properties. We show that one of these properties is provably required to recognize PARITY at arbitrary lengths using an SSM (Theorem 2). (c) Modeling $a^n b^n$: the matrix $A$ adds the previous hidden state to the update, and depending on whether the input symbol requires counting up or down, matrix $B$ is set to 1 or $-1$, thus making the SSM simulate a counter (Theorem 5)

## 3 Theoretical Results

### 3.1 Length-Generalizing Representations for Flip-Flop State Tracking

Flip Flop languages [Liu et al., 2023a] are a simple instance of state tracking defined in terms of *write*, *read*, and *ignore* instructions. Each *write* instruction comes with a piece of information; whenever a *read* instruction is encountered, the information written by the last *write* instruction is recalled. Formally, $\mathcal{L}_{FF}$ is the set of finite strings **x** over $\Sigma = \{\mathtt{r}, \mathtt{w}, \mathtt{i}, 0, 1\}$, where $x_1, x_3, \cdots \in \{\mathtt{r}, \mathtt{w}, \mathtt{i}\}$, $x_2, x_4, \cdots \in \{0, 1\}$, and where the bit following any $\mathtt{r}$ matches the bit following the last preceding occurrence of $\mathtt{w}$. Liu et al. [2023b] show that the Flip Flop language, as an abstraction, is a fundamental ingredient of many long-range reasoning settings. It can be represented with a small finite-state-automaton, and LSTMs learn $\mathcal{L}_{FF}$ well [Liu et al., 2023a]. Transformers can in principle represent it [Liu et al., 2023b,a], though known constructions are not inherently length-generalizing, a fact confirmed empirically; intuitively, this may happen because attention heads aggregate information in a commutative manner, and reliably attending to the last *write* instruction requires strong position dependence in the attention weights. SSMs, similar to traditional RNNs can easily represent Flip Flop at arbitrary input lengths and thus **avoid a failure mode of self attention**:

**Theorem 1.** *There is a two-layer SSM that predictively models $\mathcal{L}_{FF}$ at all lengths, at finite precision.*

In the construction (Figure 3), the first layer records the last instruction token, achieved in (1) by setting $A(e(\mathtt{r})) = A(e(\mathtt{w})) = A(e(\mathtt{i})) = 0$, and $A(e(0) = A(e(1)) = 1$, and setting $B(e(0)) = B(e(1)) = 0$. Additional dimensions forward the current token to $h_t^{(1)}$. In the output of the first layer $z_t^{(1)}$, whenever the input is 0 or 1, the model now has access both to the current token $w_t$ and the preceding token $w_{t-1}$, which must have been an instruction. Based on this information, the model can set the gate to overwrite the state $h_{t-1}^{(2)}$ with the current input token when the preceding token was $\mathtt{w}$, and pass along the state $h_{t-1}^{(2)}$ unaltered otherwise. This, together with $z_t^{(1)}$, is sufficient for always identifying the legal next symbols in $\mathcal{L}_{FF}$. The formal proof is in Appendix B.1.

### 3.2 Difficulty of PARITY

PARITY, the language of bitstrings with an even number of ones, is recognized by a finite-state automaton with 2 states, and is straightforwardly encoded into a traditional RNN, even a linear one, with finite precision. It is in principle expressible for transformers [Chiang and Cholak, 2022], but is empirically hard for transformers to learn [Bhattamishra et al., 2020, Deletang et al., 2022], as it can provably only be represented in sharp minima [Hahn and Rofin, 2024]. A sufficiently general SSM could easily recognize it at $d = 1$ by setting $h_0 = 1$, $A(e_1) = -1$, $A(e_0) = 0$, $B \equiv 0$, so that the sign of the single entry of $h_t$ indicates the parity (Figure 2). Such an SSM would need to be non-time-invariant and require negative or complex gate values; i.e., satisfy neither TIME-INVARIANT

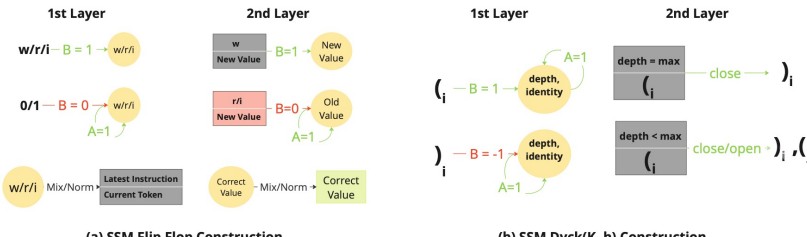

Figure 3: (a) Construction for Flip-Flop (Theorem 1): The first layer stores instruction bits to the hidden state, while data bits are forwarded to the output. Hence, the output always contains both the latest instruction and the associated data bit. In the second layer, if the instruction bit is w, the corresponding data bit is written to the hidden state, else the old value persists. This allows the model to consistently output the correct data bit. (b) Construction for Dyck(K, h) (Theorem 6): The first layer tracks the depth by counting up for each opening bracket, and down for each closing bracket. The second layer builds on the Flip-Flop construction to find the last opening bracket at the current depth; the next symbol can be either the matching closing bracket or – if the maximum depth has not been reached – an arbitrary opening bracket.

nor NONNEGATIVE. Thus, these design choices necessitated by optimization, limit the power of an SSM in emulating finite-state-automata, establishing an **even stronger separation** between existing SSM variants and traditional RNNs than the circuit complexity arguments in Merrill et al. [2024]

**Theorem 2.** *No SSM satisfying* NONNEGATIVE *can recognize PARITY at arbitrary input lengths with finite precision. In particular, this applies to Mamba.*

The proof is in Appendix B.2; it examines inputs of the form $1^N$ and shows that the activations $z_N$ converge as $N \to \infty$, and thus cannot reliably encode the parity of $N$. It should be noted that we require the layer-wise operations used in the SSM to be either linear or based on the GLU or SwiGLU activation functions, as seen for instance in Mamba (Remark 15). As we show in Theorem 13, the same result holds even for SSMs evading NONNEGATIVE when they are TIME-INVARIANT, at least when the coefficients have rational angles in the complex planes. All extant SSMs we surveyed (Appendix, Section A) satisfy either NONNEGATIVE or TIME-INVARIANT. Hypothetical SSMs evading both NONNEGATIVE and TIME-INVARIANT would be strictly stronger and can represent not only PARITY, but *all* regular languages known to be in $TC^0$ (Theorem 22).

### 3.3 Exact characterization of Regular Languages modeled by SSMs

We combine Theorems 1 and 2 to derive an exact characterizations of the regular languages that modern non-time-invariant SSMs such as Mamba can recognize or predictively model – the two notions coincide here – in the finite-precision setting. The key insight is that $\mathcal{L}_{FF}$ and PARITY are fundamental building blocks of two classes of regular languages: *star-free languages* and their complement, *non-star-free languages* [Schützenberger, 1965, McNaughton and Papert, 1971]:

**Definition 3.** *A regular language is* star-free *if it can be defined using regular expressions involving only the empty set, the empty string, individual symbols, concatenation, and Boolean combinations – avoiding the Kleene star operation.*

$\mathcal{L}_{FF}$ is star-free: there is a way to define it without Kleene star. PARITY is not star-free; any regular expression for it must involve the Kleene star. Some languages that are intuitively defined with Kleene stars may still be star-free.[2] A language is star-free if and only if it can be defined logically using only first-order quantifiers and the order relation [Schützenberger, 1965]. Also, $\mathcal{L}$ is non-star-free if and only if recognizing it involves counting modulo some finite integer $K$ [McNaughton and Papert, 1971]; Modern non-time-invariant SSMs such as Mamba cannot perform modulo counting, but they can model *all* star-free languages:

**Theorem 4.** *Let $\mathcal{L}$ be a regular language. The following are equivalent:*

---

[2]For example, $(01)^*$ is star free. It is the union of $\varepsilon$ with the intersection of $0\Sigma^*$, $\Sigma^*1$, with the complements of $\Sigma^*00\Sigma^*$ and $\Sigma^*11\Sigma^*$.

1. *There is an SSM satisfying* NONNEGATIVE *that predictively models* $\mathcal{L}$ *at all input lengths, at finite precision*

   2. $\mathcal{L}$ *is star-free.*

The proof in Appendix B.3 uses the Krohn-Rhodes theorem [Krohn and Rhodes, 1965] to reduce all star-free languages to flip flop-like state tracking. Importantly, there are well-known constructive criteria for deciding whether a given automaton defines a star-free language [Schützenberger, 1965]; hence, we have a *decidable criterion* for the finite-state tracking problems that such SSMs satisfying NONNEGATIVE can solve.

This is much simpler than the situation for transformers, where an exact characterization of their power within the regular languages is complicated: Angluin et al. [2023] show that a certain formal abstraction of transformers (masked unique hard attention) also recognizes exactly the star-free languages, but constructions of realistic transformers via Krohn-Rhodes in Liu et al. [2023b] do not inherently length generalize. Both theoretical [Huang et al., 2024] and empirical research indicate difficulties in generalizing even for some simple star-free languages [Bhattamishra et al., 2020, Liu et al., 2023a]. Known length-generalizing constructions are limited to very simple subclasses such as the piecewise testable languages [Yang and Chiang, 2024]. In contrast, for SSMs we have a single model per language, at finite precision and for arbitrarily long inputs. Thus, we expect that the SSM architecture confers an advantage in star-free state tracking problems when compared to transformers – a prediction we will find supported experimentally (Figure 5).

### 3.4  SSMs can perform unbounded counting

Having characterized the regular languages modeled by SSMs, we now consider languages requiring unbounded counting [Fischer et al., 1968b], specifically, languages recognized by keeping track of one or more counters, where each character causes a specific increment or decrement to each counter [Krebs et al., 2015, Hahn et al., 2015, Weiss et al., 2018, Kutrib et al., 2021]. A prime example is the Dyck-1 language of well-formed strings over "(" and ")"; here a counter is incremented (decremented) whenever an opening (closing) bracket is encountered; a string is well-formed if and only if the counter is 0 at the end of the string. Some other relevant formal languages are Shuffle-Dyck-$k$ (the shuffles of multiple Dyck-1 languages), $a^n b^n$ – here, $a$ increments the counter and $b$ decrements it, and $a^n b^n c^n$ – here, there are two counters, one keeping track of $a^n b^n$ and one of $b^n c^n$ (See Appendix C.2). Such counter languages are fundamental as basic context-free (Dyck-1, $a^n b^n$) or context-sensitive (e.g., $a^n b^n c^n$) languages [Hopcroft et al., 2001], and have been the subject of studies of both transformers [Bhattamishra et al., 2020] and RNNs [Weiss et al., 2018].

**Theorem 5.** *The languages Dyck-1, Shuffle-Dyck-k, n-ary Boolean Expressions, $a^n b^n$, $a^n b^n c^n$, and $a^n b^n c^n d^n$, (defined in Appendix C.2) can each be predictively modeled by an SSM.*

The proof is in Appendix B.4. Intuitively (Figure 2), an SSM can directly implement the required counters by setting $A \equiv 1$ and by defining $B(e_\sigma)$ to be the increment or decrement cased by $\sigma$. In modeling such languages, SSMs pattern with both transformers [Bhattamishra et al., 2020] and LSTMs [Weiss et al., 2018].

It may seem counterintuitive that NONNEGATIVE SSMs can perform unbounded counting but (by Theorem 2) not modular counting—the latter would seem to just require reading out the value of an unbounded counter. What is key is that, even though $h_t$ can encode unbounded counts, reading out the modular value of an unbounded integer is a formidable problem for typical neural network nonlinearities, in particular when the information has been pushed through normalization (2).

We should note that there is a qualitative difference between this result and the preceding positive results about finite-state languages (Theorems 1 and 4), in that the construction in Theorem 5 uses unboundedly large entries in the state $h_t$, whereas Theorems 1 and 4 use bounded values at finite precision. Indeed, we will find better length generalization in the finite-state case (Figure 5).

A consequence of Theorem 5 is that SSMs can recognize some languages transcending the context-free languages, as $a^n b^n c^n$ is not context-free. A second application of the theorem, of great linguistic interest, is to bounded hierarchical structure, as we discuss next.

### 3.5 Bounded Hierarchical Structure without Stacks

It is generally agreed that hierarchical structure is a key aspect of language, and comprehending language at a human-like level requires the computational ability to process such structures [Chomsky and Schützenberger, 1963, Linzen et al., 2016, Everaert et al., 2015]. The fundamental data structure for the same is a stack, where information is stored and removed as one traverses to higher and lower levels of hierarchical embedding [Hopcroft et al., 2001]. We now show that SSMs' counting ability can offer shortcuts on languages modeling hierarchical structure, eschewing the need for a stack.

A useful abstraction of hierarchical structure as relevant to natural language is the family of Dyck languages. The bounded-depth Dyck language $Dyck_{K,h}$ with $K$ types of parentheses and depth $h$ is the language of well-bracketed strings over $(_1, )_1, \ldots, (_K, )_K$, such that the number of yet unclosed brackets never exceeds $h$ in any prefix [Hewitt et al., 2020, Yao et al., 2021b]. The Chomsky-Schützenberger theorem [Chomsky and Schützenberger, 1963] asserts that any context-free language can be expressed as a homomorphic image of the intersection between a Dyck language and a regular language. Specifically, the Dyck language in question refers to the classical unbounded-depth Dyck language, where $h \to \infty$, underscoring its fundamental role as the structural backbone of context-free languages. Bounding the depth reflects the fact that deep embedding is rare in natural language [Karlsson, 2007, Blasi et al., 2019]. Prior work has found that two-layer transformers [Yao et al., 2021a] and traditional RNNs [Hewitt et al., 2020, Bhattamishra et al., 2020] both model all $Dyck_{K,h}$ languages. The same turns out to hold for SSMs:

**Theorem 6.** *There is a two-layer SSM with $d = O(h \log K)$ that predictively models $Dyck_{K,h}$ at all input lengths, at finite precision.*

The proof is in Appendix B.5. Intuitively (Figure 3), the first layer records the depth of each parenthesis using the ideas from Theorem 5, and the second layer keeps track of the last open bracket at each depth using Theorem 1. We note that, since $Dyck_{K,h}$ is star-free, Theorem 4 already guarantees the existence of representing SSMs, but the depth and width guaranteed by Theorem 6 is likely to be much better than what would be obtained by a black-box application of Theorem 4: As Hewitt et al. [2020] show, $h \log K$ units is optimal up to constants and is attained by traditional RNNs and LSTMs. The SSM construction is very different from that of Hewitt et al. [2020] for traditional RNNs (both simple RNNs and LSTMs), which directly simulates a stack. Our construction is similar to the transformer construction in Theorem 4.2 in Yao et al. [2021a], which however has to rely on specific positional encodings, unlike the SSM construction. This highlights that stacks are not the only way of simulating bounded hierarchical structure in recurrent architectures, and non-stack-based strategies can even attain the same optimal

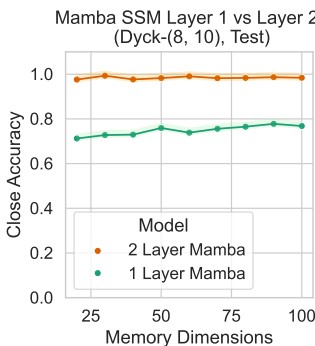

Figure 4: As predicted by Theorem 6, Mamba with 2 layers can model Dyck(K, h). Results for test set with strings of length $700 \le n \le 1400$.

scaling of hidden units. Probing whether such stack-free shortcuts are learned by SSM-based LLMs is an exciting problem for future research.

## 4 Experiments

We have derived a fine-grained theoretical characterization of expressiveness strengths and limitations of SSMs. We now show that our positive results can be instantiated and learned in a realistic SSM implementation, by evaluating a recent highly successful SSM, Mamba [Gu and Dao, 2023].

**FlipFlop** We empirically instantiate Theorem 1 using the dataset of Liu et al. [2023a], reflecting the language $\mathcal{L}_{FF}$ as defined in Section 3.1. Matching Figure 2 in Liu et al. [2023a], we evaluated both on in-distribution data, and on out-of-distribution data where the distance between read and write instructions tended to be larger. We evaluate for predicting the bits following $r$ instructions[3],

---

[3]Predictive modeling is trivial at other positions, as only the input symbols need to be considered there.

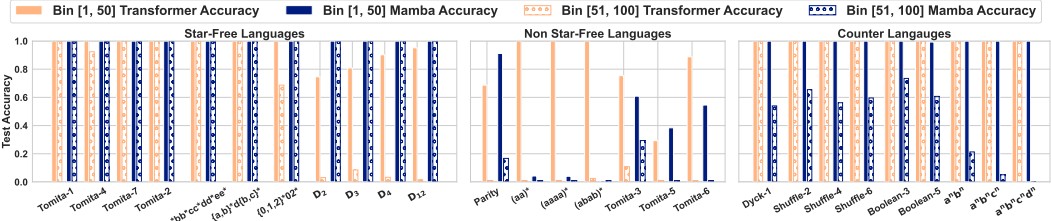

Figure 5: Results on 27 formal languages, comparing our Mamba results (blue) with transformer results reported by Bhattamishra et al. [2020] (orange), on in-distribution lengths (solid) and out-of-distribution lengths (dotted). As predicted by Theorem 4, Mamba performs strongly on star-free languages, and even shows perfect length generalization. Again as predicted by Theorem 4, it performs poorly on non-star-free languages. Results for transformers from Bhattamishra et al. [2020] are mixed. Mamba also succeeds on learning the counter languages from Theorem 5, showing perfect accuracy at in-distribution lengths at in-distribution lengths, but length generalization lags behind transformers.

matching the "deterministic/clean" mode of Liu et al. [2023a], and considered predictions to be correct only if all predictions within a sequence were correct. (Further details in Appendix D.2). A small one-layer[4] Mamba model converged to 0 error in both validation sets after $\sim$ 1400 steps (Figure 6), compared to 500 steps for an LSTM reported by Liu et al. [2023a]. In contrast, Liu et al. [2023a] found that transformers kept making occasional mistakes despite training for 10K steps.

**Test Suite from Bhattamishra et al. [2020]** To test our theoretical results on regular and counter languages (Theorems 2, 4, 5), we test Mamba on 27 formal languages, including 18 regular languages and 9 counter languages, based on a prior study comparing transformers and RNNs [Bhattamishra et al., 2020]. The regular languages include a popular benchmark [Tomita, 1982] and various regular expressions; 11 are star-free. The counter languages include the languages covered by Theorem 5. (Definitions in Appendix C). We chose this test suite as it precisely covers Theorems 4 and 5, and we have proven (in)expressibility results for each language in the set.

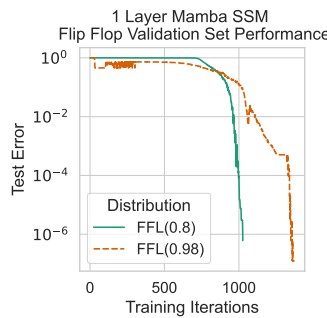

Figure 6: Test error on the validation set for $\mathcal{L}_{FF}$, following Liu et al. [2023a]. Mamba shows near-zero test error in both In- (green) and Out-of-distribution (orange) settings, consistent with Theorem 1, and avoids the failure seen in transformers [Liu et al., 2023a]

Following Bhattamishra et al. [2020], we trained the model for predictive modeling, i.e., at each step, the model outputs a label indicating the set of possible next characters (3), including EOS when required. Following Bhattamishra et al. [2020], we count the model's response on an input string as correct if and only if predictive modelling was successful at *all* positions in the input. Such a evaluation setup makes random baselines low, where a random predictor would have an accuracy exponentially small in $N$ in each of the $N$ positions. Training inputs have length in [1,50]; the model is evaluated on held-out bins with length [1,50] and [51,100]. Further experimental details are in Appendix D.1.

We show our Mamba results, together with Transformer results reported by Bhattamishra et al. [2020], in Figure 5. LSTMs perform perfectly on all languages, and are thus not shown. In a striking confirmation of Theorem 4, Mamba learns all star-free languages with strong length generalization but performs poorly on non-star-free languages. Transformers show more mixed results, often failing to length-generalize even on star-free languages. Consistent with Theorem 5 , Mamba, like Transformers, learns counter languages but struggles more with length generalization. The differences in Mamba's performance between star-free and counter languages may stem from the fact

---

[4]Theorem 1 constructs a *two*-layer SSM. We hypothesize that Mamba uses its local convolution (Remark 19) to replace the lower layer from the construction in Theorem 1.

that the construction for the former class (Theorem 4) is able to use finite precision and bounded state values at arbitrary input lengths, while the latter (Theorem 5) uses unbounded state values.

**Bounded Hierarchical Structure**  To test Theorem 6, we recreate the experimental setup from Yao et al. [2021b]. Matching their Figure 4, we trained Mamba to predictively model $Dyck_{K,h}$ at $K = 8$ and $h = 10$. The training and the validation set contained samples of length $\leq 700$, while the test set contained samples of length $700 \leq n \leq 1400$. Yao et al. [2021b] found both transformers and LSTMs achieved strong performance on this setup. We provide further details in Appendix D.3. Recall that Theorem 6 shows that two-layer SSMs can predictively model $Dyck_{K,h}$. We trained Mamba with 1 or 2 layers and varying dimensionality, finding that two layers can achieve essentially perfect performance across model sizes, even on the test set (Figure 4 and 7).

## 5   Discussion

**Related Work**  Our work belongs to an incipient line of research into the expressiveness of SSMs [Jelassi et al., 2024, Merrill et al., 2024]. It is closely related to a long string of work studying the expressive capacity of neural sequence models, which has so far focused on recurrent networks [e.g. Siegelman and Sontag, 1995, Bhattamishra et al., 2020, Hewitt et al., 2020] and, more recently, self attention [e.g. Chiang et al., 2023, Merrill and Sabharwal, 2023, Strobl et al., 2024]. A second link is to the classical and long-standing study of linear dynamical systems and control theory [Kalman, 1960]. For instance, Theorem 2 relies the asymptotic convergence of an SSM on certain inputs, establishing a link to the asymptotics of linear systems [e.g. Phillips and Solo, 1992].

**Take-Aways**  While theoretical in nature, our results have several actionable implications for SSM and LLM research, informing the rapidly growing research on SSM-based LLMs. *First*, encouragingly, SSMs can keep track of bounded hierarchical structure with optimal memory even without explicitly implementing a stack (Theorem 6), suggesting that simple diagonal linear state updates may be sufficiently powerful for modeling the hierarchical structure of language. *Second*, SSMs resolve a basic failure mode of self-attention in flip-flop state tracking while being parallellizable (Theorem 1). Overall, SSMs and attention have overlapping but distinct strengths. This lends support to the development of hybrid architectures interleaving SSM and attention layers, as instantiated very recently by Jamba [Lieber et al., 2024]. *Third*, nonnegative gates as obtained by exponential or sigmoid parameterizations provably restrict expressive capacity, even in non-time-invariant SSMs (Theorem 2). While Gu and Dao [2023] found no evidence that complex-valued paramerizations improved over real-valued ones in the language modality, our results suggest revisiting this question, at least for tasks where periodic state-tracking abilities may be important. *Fourth*, while exactly characterizing the capacity of transformers has proven difficult even in the finite-state case, Theorem 4 provides a decidable characterization of the regular languages – equivalently, finite-state tracking problems – that SSMs such as Mamba can model. Such decidable characterizations may make it easier to theoretically predict abilities and anticipate failures of LLMs; exploring the implications of this characterization in more realistic setups is an exciting direction for future research.

**Limitations**  The main limitation of our theoretical results is that they focus on in-principle expressiveness, and do not directly make statements about learning and generalization. Future work could address this, for example, by examining whether our constructions result in reasonably flat minima, or by studying gradient flow dynamics. While we empirically verified that our positive results can indeed be instantiated, in a learnable manner, in one realistic SSM implementation, implementational differences might still result in practical differences between implementations. Studying the role of such implementational differences is an interesting problem for future work; we have made a first step by theoretically elucidating the implications of nonnegative gate values.

## 6   Conclusion

We have studied the expressive capacity of modern state space models (SSMs), through the lens of automata and formal languages. We have shown theoretically that SSMs can express star-free languages, a range of counter languages, and bounded hierarchical structure. By providing rigorous results about the expressiveness of the SSM architecture, our results can provide guidance to work on SSM-based language models.

## Acknowledgments

We thank Mark Rofin for useful discussion about Theorem 2 and we thank anonymous reviewers for their helpful feedback. We gratefully acknowledge the insightful discussions with members of the FLaNN community which contributed to the ideas of this project. Funded by the Deutsche Forschungsgemeinschaft (DFG, German Research Foundation) – Project-ID 232722074 – SFB 1102.

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

## A Instantiations of General Framework in SSM Models

Here, we survey how (1) is instantiated in a range of SSMs. As stated in Section 2.1, we refer to SSMs where the gate $A$ does not depend on $x_t$ as *time-invariant*. An equivalent terminology is the distinction between "Weak Linear Time Invariant Convolutional Models" (i.e., time-invariant) and "Linear Time Variant Models" (i.e., non-time-invariant) in Akyürek et al. [2024].

### A.1 Non-Time-Invariant Models

Approximately simultaneously with or more recently than Gu and Dao [2023], a range of non-time-invariant SSMs have been introduced [De et al., 2024, Yang et al., 2024, Qin et al., 2024b,a]. This category also covers highly similar earlier RNN variants [Bradbury et al., 2016, Lei et al., 2018].

**Mamba** In Mamba, (2) and (3) directly map onto Eqs. (2a) and (2b) in Gu and Dao [2023]. The notation of Gu and Dao [2023] use a matrix multiplication $\overline{A}h_{t-1}$ instead of elementwise multiplication $A(x_t) \circ h_{t-1}$ in (REF), but importantly, Mamba's $\overline{A}$ is diagonal, so we can take $A(x_t)_i = \overline{A}_{ii}$. Due to exponential parameterization, its entries are nonnegative.

**Griffin** The RG-LRU layer of Griffin [De et al., 2024] uses the equation

$$h_t = \underbrace{a_t}_{A(x_t)} \circ h_{t-1} + \underbrace{\sqrt{1 - a_t^2} \circ (i_t \circ x_t)}_{B(x_t)}$$

where $a_t, i_t$ are neurally parameterized in terms of $x_t$ but not $h_{<t}$; by design, $a_t \in (0, 1)$. $\phi$ is instantiated in terms of linear transformations, GeLU, and RMSNorm (Figure 2 in De et al. [2024]). The local attention used by Griffin can be subsumed into an SSM layer (Remark 19).

**Gated Linear Attention [GLA Yang et al., 2024]** This model (Section 4.4 in Yang et al. [2024]) instantiates our framework using a recurrence of the form (1); while the state is two-dimensional in this model, the update is performed by elementwise products as in (1). The gate is obtained by applying sigmoid to a linear transformation of $x_t$; thus, its entries are in $(0, 1)$. $\phi$ is instantiated in terms of SwiGLU and LayerNorm.

**HGRN** HGRN [Qin et al., 2024b] and HGRN2 [Qin et al., 2024a] are defined by a recurrence of the form (1); the gate entries are $\in (0, 1)$ by design. $\phi$ is instantiated in terms of GLU, linear transformations, and normalization. In HGRN, the state is complex, but crucially the gate remains real-valued.

### A.2 Time-Invariant Models

Time-invariant SSMs introduced before late 2023 are surveyed by Gu and Dao [2023, Appendix B], such as [Mehta et al., 2023, Sun et al., 2023, Orvieto et al., 2023]. Time-invariant SSMs have often used complex-valued states and gates; this does not have a major impact on our results: First, as complex-valued SSMs subsume real-valued ones, our positive results carry over. Second, our negative result about PARITY is affected by this distinction and requires a separate argument, see Theorem 13.

Note also that $Ah_{t-1}$ is often described as a general matrix multiplication, but $A$ is diagonalizable (e.g. Lemma 3.2 in Gu et al. [2021]; Sun et al. [2023] for RetNet), which —even though implementation may be based on non-diagonalized representations [Gu et al., 2021]—renders the model equivalent to one where $A$ is diagonal from the start. This equivalence is shown as Lemma 3.1 in Gu et al. [2021].

## B Formal Definitions and Proofs

### B.1 Flip Flop

We begin by introducing key notions of automata theory. References for automata theory include Eilenberg [1974], Hopcroft et al. [2001], Sakarovitch [2009]. We will provide those key notions that

are necessary to prove our results. We will focus on *deterministic* finite-state-automata (DFA), and simply refer to them as *finite-state-automata*.[5] First,

**Definition 7.** *A (deterministic) finite-state-automaton $\mathcal{A}$ consists of:*

- *a finite alphabet $\Sigma$*

- *a finite state set $Q$*

- *a starting state $q_0 \in Q$*

- *a transition function $u : Q \times \Sigma \to Q$*

*We extend $u$ to a map $u : Q \times \Sigma^* \to Q$ by setting:*

$$u(q, \varepsilon) = q$$
$$u(q, w_{1\ldots i+1}) = u(u(q, w_{1\ldots i}), w_{i+1})$$

*where $\varepsilon$ is the empty word.*

*Intuitively, $u(q_0, \mathbf{w})$ is the state that $\mathcal{A}$ is in after reading $\mathbf{w}$.*

*The automaton recognizes a language $L \subseteq \Sigma^*$ if there is a recognizing set $R \subseteq Q$ such that*

$$L := \{w : u(q_0, \mathbf{w}) \in R\} \tag{4}$$

Kleene's Theorem [Kleene, 1951] asserts that a language $L \subseteq \Sigma^*$ is regular (i.e., defined by a regular expression) if and only if it is recognized by some finite-state automaton.

A very fundamental automaton underlying Flip Flop is:

**Definition 8.** *A set-reset automaton is a finite-state-automaton where $(Q \setminus \{q_0\}) \subseteq \Sigma$ and*

$$u(q, \sigma) = \begin{cases} q & \text{if } \sigma \notin Q \\ \sigma & \text{else} \end{cases} \tag{5}$$

Intuitively, such an automaton keeps recording the last seen symbol from a designated set $Q \subseteq \Sigma$. Such an automaton is easily simulated with a single non-time-invariant SSM layer:

**Lemma 9.** *Let $\mathcal{A} = \langle \Sigma, Q, q_0, u \rangle$ by a set-reset automaton. Then there is a single-layer SSM with finite precision and width $d = 1 + \log Q$ that maps each $w_{1\ldots T} \in \Sigma^*$ to the state sequence $u(q_0, w_1), u(q_0, w_{12}), \ldots, u(q_0, w_{1\ldots T}) \in Q^T$.*

*Formally, there is an injective map $V : Q \to \mathbb{R}^d$ such that $\rho(z_t) = V(u(q_0, w_{1\ldots t}))$ for $t = 1, \ldots, T$.*

*Proof.* Let $B(\sigma) \in \mathbb{R}^{\log |Q|}$ be a binary encoding if $\sigma \in Q$, and $\mathbf{0} \in \mathbb{R}^{\log |Q|}$ else. Take $h_0 = B(q_0)$. Let $A(\sigma) = \mathbf{0}$ if $\sigma \in Q$ and $A(\sigma) = \mathbf{1}$ else. After processing a string, the state $h_t$ is $B(\sigma)$ where $\sigma$ is the last symbol in $Q$ that has occurred if any has, and $B(q_0)$ otherwise. Coming to (2, in order to avoid division by zero when normalizing if no element of $Q$ has been read, we add a dummy dimension to $h_t$ whose value is always 1. We take $\text{Mix}_1, \text{Mix}_2$ to be the identity. Note that, even though normalization will affect the numerical values, the binary encoding of $\sigma \in Q$ can still be read out with finite precision, as $1 \leq \|h_t\|_2 \leq \sqrt{1 + \log |Q|}$, and thus nonzero entries will remain bounded away from zero. $\square$

**Theorem 10** (Restated from Theorem 1)**.** *There is a two-layer SSM that predictively models $\mathcal{L}_{FF}$ at all lengths, at finite precision.*

*Proof.* In the first layer, we use Lemma 9 to simulate a set-reset automaton over the input alphabet $\Sigma_1 = \{w, r, i, 0, 1\}$ where $Q_1 = \Sigma_1 \cup \{q_0\}$. This layer outputs at each position whether the last instruction was write, read, or ignore. The layer additionally, at each position, forwards the input symbol using additional dimensions. Formally, at the first layer, $\rho(h_t)$ allows us to read out the input symbols $x_{t-1}, x_t \in \Sigma$.

---

[5]A closely related notion is the *semiautomaton*, which is the notion considered in the closely related work Liu et al. [2023b]. Semiautomata lack a fixed start state $q_0$. We include $q_0$, but this difference is not substantial for our formal results.

In the second layer, we again use Lemma 9 to simulate a set-reset automaton over an extended alphabet $\Sigma_2 := \Sigma_1 \times \Sigma_1$, where the first component indicates the input symbol $x_t$ and where the second component indicates $x_{t-1}$. In this set-reset automaton, $Q_2$ contains, besides a start state $q_0$, those elements of $\Sigma_2$ whose second entry is $w$. The second layer thus keeps track of the input bit $b \in \{0,1\}$ following the last write instruction. It additionally forwards the input symbol $x_t$ using additional dimensions.

The second layer, via $\rho$, then predicts the possible next symbols on the basis of this information: If $x_t \in \{0,1\}$, any instruction in $\{w,r,i\}$ is possible. If $x_t \in \{w,i\}$, any bit in $\{0,1\}$ is possible. If $x_t = r$, the bit stored after the last write instruction is possible; if no write instruction has appeared (hence, the second automaton is still in its start state), any bit in $\{0,1\}$ is possible. $\qquad\square$

## B.2 Difficulty of Representing PARITY

**Definition 11.** *PARITY is the regular language over $\Sigma = \{0,1\}$ of strings where the number of ones is even. As a regular expression, PARITY is $(0^*10^*10^*)^*$.*

**Theorem 12** (Restated from Theorem 2). *No SSM satisfying* NONNEGATIVE *can recognize PARITY at arbitrary input lengths with finite precision.*

*Proof.* We consider an SSM with multiple layers, and indicate the layer in superscript: $h_t^{(1)}, \ldots, h_t^{(L)}$. We write $z_t^{(0)}$ for the input token embedding $e(w_t)$. Consider a SSM processing the word $1^t$, for $t \to \infty$. We show, by induction over the number of layers, the following claim:

(†) *Each entry of $z_t^{(k)}$ converges to a value bounded, in absolute value, by a constant.*

By the assumption of finite precision, convergence automatically leads to the entries becoming ultimately constant. Once we have shown this, we know that $z_t^{(L)}$ is constant when $t$ is sufficiently large; thus, the parity of the string $1^t$ cannot be read out from $z_t^{(L)}$. As a consequence, the SSM cannot recognize PARITY. Indeed, we have shwon the stronger claim that the language $(11)^*$ – the language of even-length strings over one symbol – is not recognized by an SSM; we will use this stronger statement in Corollary 14.

We proceed to proving (†). The claim (†) is trivially true at $k = 0$, as the input token is always the same and we defined $z_t^{(0)} := e(w_t)$. Now consider $k > 0$. By hypothesis, the activations are given as

$$h_t^{(k)} = A(x_t) \circ h_{t-1}^{(k)} + B(x_t) \tag{6}$$

where $A(x_t), B(x_t)$ are constant $\alpha := A(x_t)$, $\beta := B(x_t)$ when $t > T_0$, for some $T_0 > 0$. The solution of the recurrence for $t > T_0$ is

$$h_t = \alpha^{t-T_0}\left(h_{T_0} + \frac{\beta}{\alpha-1}\right) + \frac{\beta}{1-\alpha} \tag{7}$$

Each dimension $j = 1, \ldots, d$ of this vector can be constant (if $(h_{T_0})_j + \frac{\beta_j}{\alpha_j-1} = 0$), diverge exponentially ($\alpha_j > 1$), converge exponentially ($\alpha_j < 1$) or diverge linearly ($\alpha_j = 1$).

We next need to show that $z_t = \mathrm{Mix}_2(\mathrm{Norm}(\mathrm{Mix}_1(h_t, x_t)))$ converges.

First, consider the effect of applying a linear transformation to the state $h_t$. Each entry of the result will be some linear combination

$$u_t = \lambda_1(h_t)_1 + \cdots + \lambda_d(h_t)_d \tag{8}$$

If each $\alpha_j < 1$, then $u_t$ converges. If some $|\alpha_j| \geq 1$, there may be some cancellation if $\alpha_i = \alpha_j$ for some $i \neq j$; cancellation can only lead to full erasure of the relevant terms or to a remaining term with the same exponent. In conclusion, each entry $u_t$ will again either converge to a finite value or diverge towards $\pm\infty$.

We now need to understand the behavior of $\mathrm{Mix}_1(h_t, x_t)$. Recall that, based on our survey (Appendix A), we allowed it to contain linear, GLU [Dauphin et al., 2017], and SwiGLU [Shazeer, 2020] components. If $\mathrm{Mix}_1(h_t, x_t)$ implements a linear transformation only, each entry likewise may

converge, diverge linearly, or diverge exponentially. We note that—if $\sigma$ is the sigmoid function—$\sigma(u_t)$ always converges, as $\sigma$ simply saturates to 0 or 1 if $u_t$ diverges. Hence, if $\mathrm{Mix}_1(h_t, x_t)$ implements GLU, each entry likewise may converge, diverge linearly, or diverge exponentially. Finally, if $\mathrm{Mix}_1(h_t, x_t)$ implements SwiGLU, each entry of the result will be a product of a linear combination of the form $u_t$, and $Swish_\beta$ applied to another such linear combination. Depending on the behavior of these two $u_t$-like terms, the outcome will behave as a product of sequences that may converge exponentially, diverge exponentially, or diverge linearly – e.g., the outcome may also diverge quadratically, or converge as $n\alpha^{-n}$, etc.

If all dimensions of $\mathrm{Mix}_1(h_t, x_t)$ converge, then $\mathrm{Norm}(\mathrm{Mix}_1(h_t, x_t))$ will also converge to a scaled version of $\frac{\beta_i}{1-\alpha_i}$, scaled by a bounded factor as $\beta_i \neq 0$. Now assume some dimensions of $\mathrm{Mix}_1(h_t, x_t)$ do not converge; in this case, for any two dimensions $i, j$, either their ratio will converge to a constant, or converge to 0 or $\pm\infty$. After applying $\mathrm{Norm}(\cdot)$, the entries asymptotically dominating the others will converge to a finite value bounded, in absolute value, by 1; the others will converge to zero.

In conclusion, we have found that each entry of $\mathrm{Norm}(\mathrm{Mix}_1(h_t, x_t))$ converges to some number bounded, in absolute value, by 1. As $\mathrm{Mix}_2$ is continuous, each entry of $z_t$ likewise converges, with a bound depending on the Lipschitz constant of $\mathrm{Mix}_2$. □

We next show the result, referenced in the main paper text after Theorem 2, about time-invariant SSMs with complex-valued gates:

**Theorem 13.** TIME-INVARIANT *SSMs cannot recognize PARITY with finite precision at arbitrary input lengths, even with complex-valued gates, as long as each entry in each A has a rational angle in the complex plane.*

Here, by a *rational angle*, we refer to an angle that is a rational number when expressed in degrees; such angles are rational multiples of $2\pi$ when expressed in radians. As the rational angles are dense in the reals, one expects that even if some irrational angles permitted modeling PARITY, such solutions would be very hard to find – in particular given that irrational numbers are not exactly represented in finite precision.

*Proof.* By assumption, any $A_j \in \mathbb{C}$ in any layer can be written as

$$A_j = r_j \exp(2\pi i q_j) \tag{9}$$

where $q_j \in [0, 1]$ is rational and $r_j \geq 0$ is real – here, $2\pi q_j$ is known as the *argument* of $A_j$; it describes the angle of $A_j$ in the complex plane in radians. Correspondingly, the angle in degrees is described by $q_j \cdot 360°$; this is rational if and only if $q_j$ is.

As a time-invariant SSM has a finite number of such values $A_j$, across all its layers, we can select a positive integer $W$ such that $Wq_j \in \mathbb{N}$ for each $j$, in each layer. Importantly, $(A_j)^W = (r_j)^W \in \mathbb{R}$.

Now consider the action of any layer of the SSM on an input sequence of the form $A_T = (10^{W-1})^T$.

The claim is that, for each $i = 1, \ldots, W$, the sequence

$$z_{tW+i}^{(k)} \tag{10}$$

converges as $t \to \infty$. As in the proof of Theorem 2, in the finite-precision setting, converge entails that the sequence becomes ultimately stationary. Note that the parity of $A_T$ equals the parity of $T$; hence, it is impossible to read out the parity from $z_{TW}^{(k)}$ when $T$ is large.

Now consider, suppressing the index for the dimension in $1, \ldots, d$:

$$
\begin{aligned}
h_{tW}^{(k)} &= \sum_{i=1}^{tW} A^{tW-i} B(z_i^{(k-1)}) \\
&= \sum_{i=1}^{tW} A^{tW-i} B(z_i^{(k-1)}) \\
&= \sum_{s=1}^{t} \sum_{j=sW}^{(s+1)W-1} A^{tW-j} B(z_{sW+j}^{(k-1)}) \\
&= \sum_{s=1}^{t} \sum_{j=0}^{W-1} A^{(t-s)W-j} B(z_{sW+j}^{(k-1)}) \\
&= \sum_{s=1}^{t} \sum_{j=0}^{W-1} (r \exp(2\pi i q))^{(t-s)W-j} B(z_{sW+j}^{(k-1)}) \\
&= \sum_{s=1}^{t} \sum_{j=0}^{W-1} r^{(t-s)W-j} \exp(-2\pi i j q) B(z_{sW+j}^{(k-1)}) \\
&= \sum_{j=0}^{W-1} \exp(-2\pi i j q) \sum_{s=1}^{t} r^{(t-s)W-j} B(z_{sW+j}^{(k-1)})
\end{aligned}
$$

Separately considering summation beyond $T_0$ at which $z_{tW+j}^{(k-1)}$ has become stationary, we get

$$
= \underbrace{\left[ \sum_{j=0}^{T_0-1} \cdots \right]}_{U_1} + \left[ \underbrace{\left( \sum_{j=T_0}^{W-1} \exp(-2\pi i j q) B(z_j^{(k-1)}) r^{-j} \right)}_{U_2} \underbrace{\left( \sum_{s=1}^{t} r^{(t-s)W} \right)}_{U_3} \right]
$$

$U_1$ and $U_2$ do not depend on $t$. Intuitively, $U_2 \in \mathbb{C}$ determines a direction in the complex plane, whereas $U_3 \in \mathbb{R}$ determines a magnitude. It remains to understand $U_3$, which can be rewritten as:

$$
U_3 = \sum_{s=1}^{t} r^{(s-1)W} = r^{-W} \sum_{s=1}^{t} (r^W)^s = r^{-W} \sum_{s=0}^{t} (r^W)^s - r^{-W} = r^{-W} \begin{cases} \frac{1-(r^W)^t}{1-(r^W)} - 1 & r \neq 1 \\ s-1 & r = 1 \end{cases} \tag{11}
$$

We have now achieved a situation like in the proof of Theorem 2: $U_3$ can converge exponentially, diverge linearly, or diverge exponentially. The remainder of the proof is analogous to that proof. □

The following Corollary of Theorem 2 will be used in the proof of Theorem 4:

**Corollary 14.** *Assume* NONNEGATIVE*, SSMs with finite precision cannot recognize any non-star-free regular language.*

*Proof.* For any non-star-free regular language $\mathcal{L}$, there are words $u, v, w$ such that the membership $uv^n w \in \mathcal{L}$ is determined by the value of $n$ modulo some finite integer $k$ (depending on $\mathcal{L}$) [McNaughton and Papert, 1971]. Fix any such $u, v, w \in \Sigma^*$.

Now assume an SSM satisfying NONNEGATIVE can recognize $\mathcal{L}$ with finite precision. We can subsume the action of $u$ into the state $h_0$ by taking $h_0$, in each layer, to be the state of the SSM after reading $u$. We now have an SSM that can determine the parity of $t$ when fed a word of the form $v^t w$.

For this SSM, we want to show

(†) *When fed words of the form* $v, v^2, v^3, \ldots$, *for each* $i = 0, \ldots, |v| - 1$, *and each layer* $k = 1, \ldots, L$, *the sequence* $z_{t|v|+i}^{(k)}$ *converges as* $t \to \infty$.

As in the preceding two proofs in this section, convergence entails becoming ultimately constant in the finite-precision setting.

The claim (†) is immediate at $k = 0$.

Now at $k > 0$, we write:

$$h_{t|v|+i}^{(k)} = A(z_{t|v|+i}^{(k-1)}) \ldots A(z_{(t-1)|v|+i+1}^{(k-1)}) h_{(t-1)|v|+i}^{(k)}$$
$$+ A(z_{t|v|+i}^{(k-1)}) \ldots A(z_{(t-1)|v|+i+2}^{(k-1)}) B(z_{(t-1)|v|+i+1}^{(k-1)})$$
$$+ A(z_{t|v|+i}^{(k-1)}) \ldots A(z_{(t-1)|v|+i+3}^{(k-1)}) B(z_{(t-1)|v|+i+2}^{(k-1)})$$
$$+ \ldots$$
$$+ B(z_{(t-1)|v|+i}^{(k-1)})$$

On the RHS, as $t \to 0$, all terms except for $h_{(t-1)|v|+i}^{(k)}$ become constant by the inductive hypothesis. Hence, there are some $\alpha, \beta$ such that, for sufficiently large $t$,

$$h_{t|v|+i}^{(k)} = \alpha \circ h_{(t-1)|v|+i}^{(k)} + \beta \tag{12}$$

We are now, for each $i$, in the same situation as in the proof of Theorem 2]: each dimension of this recurrence can converge exponentially, diverge exponentially, or diverge linearly; as in that proof, it follows that $z_{t|v|+i}^{(k)}$ converges as $t \to \infty$.

We have shown (†).

We now follow up by showing that

(∗) *When fed words of the form $v^t w$, for each $i = 1, \ldots, |w|$, and each layer $k = 1, \ldots, L$, the sequence $z_{t|v|+i}^{(k)}$ converges as $t \to \infty$.*

Again, at finite precision, convergence entails that the sequences are ultimately constant. Again, (∗) is true at $k = 0$ trivially. When feeding the SSM words of the form $v^t w$, in each layer, the final state is in each layer $k$, at each $i = 1, \ldots, |w|$:

$$h_{t|v|+i}^{(k)} = A(z_{t|v|+|w|}^{(k-1)}) \ldots A(z_{t|v|+1}^{(k-1)}) h_{t|v|}^{(k)}$$
$$+ A(z_{t|v|+i}^{(k-1)}) \ldots A(z_{t|v|+2}^{(k-1)}) B(z_{t|v|+1}^{(k-1)})$$
$$+ \ldots$$
$$+ B(z_{t|v|+i}^{(k-1)})$$

By inductive hypothesis, for large $t$, there are $\psi_i, \gamma_i$ such that

$$h_{t|v|+i}^{(k)} = \psi_i \circ h_{t|v|}^{(k)} + \gamma_i$$

and, as shown before, each entry of $h_{t|v|}^{(k)}$ converges exponentially, diverges exponentially, or diverges linearly. Now, by assumption, one can read out, at finite precisiion, the parity of $t$ from

$$z_{t|v|+|w|}^{(L)} = \text{Mix}_1(\text{Norm}(\text{Mix}_2(\psi_{|w|} \circ h_{t|v|+i}^{(k)} + \gamma_{|w|})))$$

We now simply absorb the operation $X \mapsto \psi_{|w|} \circ X + \gamma_{|w|}$ into $\text{Mix}_2$, and obtain by the same arguments as in the proof of Theorem 2 that $z_{t|v|+|w|}^{(L)}$ converges as $r \to \infty$. This is a contradiction to the claim that the value of $t$ can be read out, modulo $k$, from $z_{t|v|+|w|}^{(L)}$ at finite precision. $\qquad \square$

**Remark 15.** *As outlined in our analysis, the assumptions in Theorem 2 are based on layer-wise operations that are either linear or based on the GLU or SwiGLU activation functions. This assumption is critical to the proof: one could design activation functions that make PARITY expressible.*

*Given a sequence $\mathbf{x} = x_1, \ldots, x_T$, consider the function $f(\mathbf{x}) = \frac{e^{i\pi\sum_{i=1}^{n} x_i} + 1}{2}$. This continuous function is designed to satisfy the condition that, for bit-strings $\mathbf{x}$, $f(\mathbf{x}) = 1$ if $\sum_{i=1}^{n} x_i$ is even, and $f(\mathbf{x}) = 0$ otherwise. At first glance, it seems like this function can be approximated by a cumulative sum layer in combination with a two-layer SSM to compute $f(x) = \frac{e^{i\pi x} + 1}{2}$.*

*However, this construction cannot be implemented under the condition for which we prove Theorem 2. This is because computing this function $f(x)$ inherently requires a layer-wise nonlinear operation (such as a MLP) capable of representing sine and cosine functions over arbitrarily large input values. Importantly, achieving a construction that works for any input length requires the ability to handle arbitrarily large inputs within a single operation.*

*A single GLU or SwiGLU activation function, or even a more classical MLP with ReLU or sigmoid activations, is not expected to represent sine and cosine functions over unbounded inputs. The reason for this limitation lies in the universal approximation results for feedforward networks. These results generally guarantee approximation within compact convergence on bounded sets, such as in the compactification of $\mathbb{R}$ or in $L^p$ spaces, as described in Cybenko [1989], Ito [1992], and Arora et al. [2016]. None of these results extend to uniform approximation of sine or cosine over the entire real line.*

*Recent work by van Nuland [2024] addresses the universal approximation capabilities in the space $C_b(\mathbb{R})$, which is the class of bounded continuous functions over $\mathbb{R}$. This result is particularly relevant since approximating sine and cosine functions uniformly over $\mathbb{R}$ would fall under this category. According to their Proposition 5.5, sine and cosine functions cannot be uniformly approximated using certain activation functions, limiting the feasibility of approximating $f(x) = \frac{e^{i\pi x} + 1}{2}$ in a typical MLP architecture.*

*Thus, it is unrealistic to expect a typical MLP, with ReLU or sigmoid activations, to implement the function $f(x) = \frac{e^{i\pi x} + 1}{2}$ uniformly for arbitrarily large inputs. Consequently, a construction based on such a function would either necessitate custom activation functions, such as periodic activations specifically designed to handle sine and cosine, or require the size of the model to scale with the input length. Either solution removes apparent contradiction with Theorem 2, as these adjustments fall outside the scope of the assumptions made in our proof.*

*Wang and Xue [2024] and Orvieto et al. [2024] provide universal approximation guarantees for SSMs, but these guarantees depend on the size of the approximating network growing with input length. This dependency is clearly stated in Proposition 3.6 and Proposition 3.9 of Wang and Xue [2024], and further emphasized by Orvieto et al. [2024]. in their Remark 2. Our results, in contrast, pertain to the existence of a single SSM capable of recognizing a formal language for any input length, independent of network size. Thus, such universal approximation results do not undermine Theorem 2.*

## B.3 Proof of Theorem 4

Our proof of Theorem 4 will rely on the algebraic theory of finite automata, specifically the cascade product and the Krohn-Rhodes Theorem [Krohn and Rhodes, 1965]. These techniques, originally developed in the 1960s, have recently been introduced to the theoretical study of transformers by Liu et al. [2023b]; we provide self-contained definitions and somewhat different notation, tailored to our proofs about state-space models. In general, we will find that the properties of state-space models allow more natural and directly length-generalizing implementations of these algebraic notions than what is possible for transfomers.

Recall the definition of a finite-state-automaton (Definition 7). Our construction will build on an important operation on automata, the cascade product [Krohn and Rhodes, 1965, Eilenberg, 1974, Ginzburg, 1968]:

**Definition 16.** *Given two automata $\mathcal{A}_1, \mathcal{A}_2$ with associated alphabets $\Sigma_1, \Sigma_2$ and state sets $Q_1, Q_2$ such that*

$$\Sigma_2 = Q_1 \times \Sigma_1, \tag{13}$$

*the cascade product $A_2 \wr A_1$ is the automaton given by*

- $\Sigma = \Sigma_1$

- $Q = Q_2 \times Q_1$

- $q_0$ *is the tuple of the starting states of* $\mathcal{A}_2, \mathcal{A}_1$

- $u(\langle q, p \rangle, \sigma) = \langle u_2(q, \langle p, \sigma \rangle), u_1(p, \sigma) \rangle$

We note that the literature usually uses "∘" for the cascade product [e.g. Eilenberg, 1974]. To avoid collision with the elementwise product "∘" (e.g., (1)), we here instead use "≀", usually used for the wreath product – a product on monoids with an effect analogous to the cascade product [Almeida, 1995].

While the formal definition is cumbersome, the intuition behind it is simple: The cascade product corresponds to first reading a word $\mathbf{w}$ with $\mathcal{A}_1$, recording the state sequence $q_0, q_1, \ldots, q_{|\mathbf{w}|} \in Q_1$ and – at each $t = 1, \ldots, |\mathbf{w}|$ – pasting the state $q_{t-1}$ together with the input symbol $w_t \in \Sigma_1$ – resulting in a word over a new alphabet $Q_1 \times \Sigma_1$, and then running $A_2$ on the resulting word. The overall state of $A_2 \wr A_1$ after reading a word is the tuple of the states reached by $A_2$ and $A_1$. Note that we write $A_2 \wr A_1$, rather than, $A_1 \wr A_2$, because the second argument of the cascade product ($A_1$) intuitively reads the input first, preprocessing it for the other automaton, $A_2$ – the cascade product can thus be viewed as a kind of function composition.

The somewhat inscrutable update rule for $u(\cdot, \cdot)$ encodes the action of $\mathcal{A}_1$ in the second component, and the action of $\mathcal{A}_2$ on the extended alphabet in the first component. There is a close analogy to the stacking of sequence models, and we will leverage this analogy to translate cascade products into multilayer SSMs. The fundamental background here is the following classical fact:

**Fact 17** (Consequence of Krohn-Rhodes Theorem [Krohn and Rhodes, 1965] and Schützenberger's Theorem [Schützenberger, 1965])**.** *Each star-free regular language is recognized by an iterated cascade product of set-reset automata,* $(\ldots(\mathcal{A}_1 \wr \ldots) \wr \mathcal{A}_{n-1}) \wr \mathcal{A}_n$, *where each* $\mathcal{A}_i$ *is a set-reset automaton.*

This result follows from the Krohn-Rhodes decomposition theorem [Krohn and Rhodes, 1965], which states that any finite-state automaton can be expressed as an iterated cascade product of simple automata, specifically finite simple groups and reset automata. Moreover, Schützenberger's Theorem [Schützenberger, 1965] characterizes star-free regular languages as those whose syntactic monoids are aperiodic, meaning they contain no nontrivial groups. Therefore, the decomposition for star-free languages involves only set-reset automata, leading to the stated cascade product structure. We now formally show that cascade products can be translated to SSM stacking. We need an auxiliary lemma, which provides a single-layer SSM that encodes the input $w_{t-1}$ in state $h_t$ – we will use it to forward information about the state of $\mathcal{A}_1$ at $t-1$ to $\mathcal{A}_2$ at $t$:

**Lemma 18.** *Let* $\Sigma$ *be an alphabet, and consider words* $w \in \Sigma^*$. *There is a one-layer SSM with* $d = 4|\Sigma|$ *such that, for* $t = 2, \ldots, |w|$, *the character* $w_{t-1}$ *can be read out from* $z_t$ *at finite precision.*

To prove Lemma 18, a first idea is to use an exponential moving average with $A = 1/2$ to encode the recent input characters in $h_t$; this effectively encodes the full history into the binary expansion of $h_t$, and in particular allows reading out the second-last input in principle. However, such a construction does not work at finite precision, because rounding may make it impossible to extract even the second-most-significant bit.[6] We avoid this problem simply by taking $A = 1/4$, effectively utilizing only every two digits in the binary expansion of $h_t$, ensuring that the second-last input can be read out at a constant margin. We now provide the formal proof:

*Proof.* We begin by showing the claim in the special case $\Sigma = \{1, 0\}$. Here, we take $d = 4$, and

$$h_0 = [0, 0, 0, 0]^T$$
$$A(e_0) = [1/4, 1/4, 0, 0]^T$$
$$A(e_1) = [1/4, 1/4, 0, 0]^T$$
$$B(e_0) = [1, 0, 1, 0]^T$$
$$B(e_1) = [0, 1, 0, 1]^T$$

---

[6]Informally, in binary, 0.0111111...111 and 0.1 are arbitrarily close.

Now we separately consider the state $h_t$ depending on the form of the prefix $w_{1...t}$ (here $w_{1...t}$ refers to first $t$ characters in the word). If $w_{1...t} = ...00$ (the last 2 characters of the prefix are 00), then

$$h_t = \begin{pmatrix} \in [1,2] \\ \in [0,1/8] \\ 1 \\ 0 \end{pmatrix} \tag{14}$$

because

$$
\begin{aligned}
h_t &= A(e_0) \circ h_{t-1} + B(e_0) \\
&= A(e_0) \circ A(e_0) \circ h_{t-2} + A(e_0) \circ B(e_0) + B(e_0) \\
&= [1/16, 1/16, 0, 0]^T \circ h_{t-2} + [1/16, 1/16, 0, 0]^T \circ [1,0,1,0]^T + [1,0,1,0]^T \\
&= \begin{pmatrix} \frac{1}{16}(h_{t-2})_1 + \frac{1}{16} + 1 \\ \frac{1}{16}(h_{t-2})_2 \\ 1 \\ 0 \end{pmatrix}
\end{aligned}
$$

By definition of $A$ and $B$, each entry in $h_{t-2}$ is in $[0,2]$; the claim (14) then follows. If $w_{1...t} = ...10$, then, by a similar calculation

$$h_t = \begin{pmatrix} \in [1,1.25] \\ \in [1/4,1/2] \\ 1 \\ 0 \end{pmatrix} \tag{15}$$

In particular, assuming $w_t = 0$, one can read off $w_{t-1}$ from $(h_t)_2$ with a margin of size 1/8. As $w_t$ is encoded in $h_t$ and due to symmetry, analogous statements hold when $w_t = 1$.

Now, for each $\sigma \in \Sigma$, we run such a one-layer SSM where 0 represents $\sigma$ and 1 represents all other characters.[7] By running these in parallel (i.e. executing these operations with the same SSM layer simultaneously, utilising the width of the SSM layer) we obtain an SSM with $d = 4|\Sigma|$ from whose states one can read out $w_{t-1}$ at finite precision. As the entries in $h_t$ are all bounded by 2, we find $\|h_t\|_2 \leq 2\sqrt{d}$ independent of $t$, and the margin is still bounded away from zero after normalization, and thus in $z_t$, where we can assume $\text{Mix}_1, \text{Mix}_2$ to be the identity. $\qquad\square$

**Remark 19.** *Some SSMs include local convolutions [e.g. Fu et al., 2023, Gu and Dao, 2023] or local attention [De et al., 2024], which aggregate information from a local window of some width $\Delta > 0$. These do not increase the expressive capacity beyond SSMs as we have defined in (1-2), as aggregation of local information can be simulated with a single SSM layer: Using the layer constructed in the proof of Lemma 18, given the state $h_t$, once one has read out $w_{t-1}$ as described in the proof, one can recover $h_{t-1}$ from $h_t$ and $x_t$; then inductively read out $w_{t-2}$ using $h_{t-1}$ and $x_{t-1}$, etc. Thus, up to any given width $\Delta > 0$, one can read out $w_{t-\Delta}, ..., w_{t-1}$ from the state $h_t$ of this layer at finite precision.*

We are now ready to translate cascade products into SSM stacking:

**Lemma 20.** *Let $\mathcal{A}_1$, $\mathcal{A}_2$ be two finite-state-automata, and assume that there are two SSMs with top-level states $z^{(L_1,1)}$ and $z^{(L_2,2)}$ that map each $\mathbf{w}$ to the state sequences under $\mathcal{A}_1$, $\mathcal{A}_2$, at finite precision.*

*Formally, on a word $\mathbf{w}$, $\rho_1(z_t^{(L_1,1)})$ and $\rho_2(z_t^{(L_2,2)})$ provide the state sequences of $\mathcal{A}_1$, $\mathcal{A}_2$.*

*Then there is an SSM with $L_1 + L_2 + 1$ layers that maps each $\mathbf{w}$ to the state sequence under $\mathcal{A}_2 \wr \mathcal{A}_2$, again at finite precision.*

We note that a conceptually related result holds for transformers [Lemma 12 in Liu et al., 2023b]. However, SSMs allow a simpler and length-independent construction, as they do not require positional encodings to implement such a construction.

---

[7]In fact, using a binary encoding of $\Sigma$, one can achieve $d = 4\log|\Sigma|$.

*Proof.* The lower layers are based on the SSM modeling $\mathcal{A}_1$. We duplicate each channel, so we now have $2d$ dimensions. We further add $d$ further dimensions that directly pass on the input embeddings, i.e., $A \equiv 0$, $B \equiv 1$, $\text{Mix}_j \equiv Id$ on these dimensions.

In the resulting SSM, $z_t^{L_1}$ indicates both $w_t$ itself, and the state reached by $\mathcal{A}_1$ after reading $w_{1...t}$. The state is redundantly indicated by two separate sets of $d$ dimensions; the character $w_t$ is indicated by $d$ further state.

Note, however, that the second automaton in the cascade product requires access to the state $q_{t-1}$ rather than $q_t$.

For this, we add a layer provided by Lemma 18, of width $4|Q|$. Additional $2d$ dimensions pass on (1) $w_t$, and (2) the state that $\mathcal{A}_1$ reaches after reading the prefix $w_{1...t}$.

We now have $L_1 + 1$ layers where $z_t^{L_1+1}$ has $2d + 4|Q|$ dimensions and indicates (1) $w_t$, (2) the state that $\mathcal{A}_1$ reaches after reading the prefix $w_{1...t}$, (3) the state that $\mathcal{A}_1$ reaches after reading the prefix $w_{1...t-1}$.

The first and third piece of information are now fed into the second SSM; the second piece is passed on in $d$ additional dimensions. As we allowed $A$ and $B$ to be arbitrary functions, we redefine these in the lowest layer of that second SSM to read out from the $4|Q|$-dimensional component indicating (3), providing the desired second-to-last state.

We have constructed an SSM with $L_1 + L_2 + 1$ layers, where $z_t^{L_1+L_2+1}$ indicates (1) $w_t$, (2) the state that $\mathcal{A}_1$ reaches after reading the prefix $w_{1...t}$, (3) the state that $\mathcal{A}_2$ reaches after reading the prefix $w_{1...t}$ pasted with the state sequence of $\mathcal{A}_1$. This information is sufficient for reading out the state sequence of $\mathcal{A}_2 \wr \mathcal{A}_1$.

Note that the number of channels may not be consistent, as it is $3d$ in the top and bottom parts, but $2d + 4|Q|$ in the middle; we simply pad to the larger dimensionality. $\quad\square$

We are now ready to show the existence of length-generalizing SSMs for any star-free state tracking problem, and conclude with the theorem:

**Theorem 21** (Restated from Theorem 4). *Let $\mathcal{L}$ be a regular language. The following are equivalent:*

1. *There is an SSM satisfying* NONNEGATIVE *that predictively models $\mathcal{L}$ at all input lengths, at finite precision*

2. *$\mathcal{L}$ is star-free.*

*Proof.* We need to show:

1. SSMs at finite precision can predictively model all star-free languages. For each language, a single SSMs is applicable at arbitrary lengths.

2. Assuming NONNEGATIVE, finite-precision SSMs cannot recognize any non-star-free regular language.

The second statement is Corollary 14; it suffices to prove the first statement.

Assume $\mathcal{L}$ is star-free. By the Krohn-Rhodes theorem, there is an automaton $\mathcal{A}$ that is a cascade product of some set-reset automata that recognizes $\mathcal{L}$. By Lemmas 9 and 20, there is an SSM that computes the state sequence of that automaton.

Now we note that, since $\mathcal{A}$ recognizes $\mathcal{L}$, the state $q$ after reading $\mathbf{w}$ is sufficient for determining the set of characters that can follow this prefix in any element of $\mathcal{L}$. For, assume otherwise, then there are words $\mathbf{w}, \mathbf{w}'$ such that $u(q_0, \mathbf{w}) = u(q_0, \mathbf{w}')$ and $\sigma \in \Sigma$ such that $\mathbf{w}\sigma\Sigma^* \cap \mathcal{L} \neq \emptyset$ but $\mathbf{w}'\sigma\Sigma^* \cap \mathcal{L} = \emptyset$; then $u(q_0, \mathbf{w}\sigma) = u(q_0, \mathbf{w}'\sigma)$ but the set $R$ (4) is reachable from $u(q_0, \mathbf{w}\sigma)$ but not $u(q_0, \mathbf{w}'\sigma)$, contradiction.

Hence, the SSM's outputs can be transformed, by composing $\rho$ with a map from states to next-character sets, to predictively model $\mathcal{L}$. $\quad\square$

**Theorem 22.** *SSMs with complex-valued coefficients evading both* NONNEGATIVE *and* TIME-INVARIANT *can represent all regular languages known to be in* $\text{TC}^0$.

We we do not use this theorem in the main paper, due to the nonexistence (as far as we know) of implemented SSMs with this property.

*Proof.* SSMs evading both NONNEGATIVE and TIME-INVARIANT can count modulo any integer $k$, using $d = 1$ and $A(e_1) = e^{2\pi i/k}$, $A(e_0) = 1$, $B \equiv 0$, $h_0 = 1$. This is a generalization of the construction for PARITY described in Section B.2, since $e^{2\pi i/2} = -1$.

The set of regular languages known to be in $\text{TC}^0$ is the set of regular languages whose syntactic monoid contains no non-solvable groups [Barrington et al., 1992]. These languages are recognized by cascade products of set-reset automata and automata perfoming modular counting [Straubing, 1994]. By the remark above, together with Lemma 9 and Lemma 20, such cascade products can be simulated by SSMs. □

### B.4 Maintaining Counters

As the first step in showing Theorem 5, we show that SSMs can maintain unbounded counters, and that one can read out the values of such counters, up to finite bounds, even at finite precision:

**Lemma 23.** *Let $C > 0$ be an integer. Let any function $u : \Sigma \to \mathbb{Z}^C$ be given. Let $L \in \mathbb{N}$. Then a one-layer SSM with finite precision can compute, at each position $i = 1, \dots, T$:*

$$\max\left(\min\left(\sum_{j=1}^{i} u(w_i), L\right), -L\right) \tag{16}$$

*in the sense that $\rho$ can read this out from $z_i^{(1)}$ with finite precision.*

*Proof.* Define $d = 2L + 1$. Define $h_0 = \mathbf{0} \in \mathbb{R}^d$. For each $x \in \Sigma$, define $A(e_x) = \mathbf{1} \in \mathbb{R}^d$ and $B(e_x)_i \in \mathbb{R}^d$ by $B(e_x)_i = u(x)$. In order to read out the state $h_t$ up to a limit $L$, we define

$$\phi(h_t, x_t) = \text{Norm}(h_t + [0, 1, -1, 2, -2, \dots, -L, L]) \tag{17}$$

By testing which entries of the result are negative or positive, one can read out the state up to $L$ even after rounding $\phi(h_t, x_t)$ to finite precision. The proof straightforwardly extends to multiple counters. □

We are ready to prove the theorem:

**Theorem 24** (Restated from Theorem 5)**.** *The languages Dyck-1, Shuffle-Dyck, n-ary Boolean Expressions, $a^n b^n$, $a^n b^n c^n$, and $a^n b^n c^n d^n$, (defined in Appendix C) can each be predictively modeled by an SSM.*

*Proof.* For each of these languages, we first define an assignment $u : \Sigma \to \mathbb{Z}^C$:



For $a^n b^n$: (here, $C=1$)

$u(a) = 1$

$u(b) = -1$

For Dyck-1: (here, $C=1$)

$u(\text{``(''}) = 1$

$u(\text{``)''}) = -1$

For Shuffle-Dyck-$k$ (here, $C = k$)

$u(\text{``}(_i\text{''}) = (0, \ldots, 0, 1, 0 \ldots 0)$     where 1 is in the $i$-th slot

$u(\text{``}{)_i}\text{''}) = (0, \ldots, 0, -1, 0 \ldots 0)$     where $-1$ is in the $i$-th slot

For $a^n b^n c^n$ : (here, $C=2$)

$u(a) = (1, 0)$

$u(b) = (-1, 1)$

$u(c) = (0, -1)$

For $a^n b^n c^n d^n$ : (here, $C=3$)

$u(a) = (1, 0, 0)$

$u(b) = (-1, 1, 0)$

$u(c) = (0, -1, 1)$

$u(d) = (0, -1, -1)$

For Boolean Expressions: (here, $C=1$)

$u(\langle VALUE \rangle) = -1$

$u(\langle n - ARY \rangle) = +n$



For each of these mappings, we use Lemma 23 at $L = 1$ to construct a one-layer SSMs that can, for each of the $C$ counters, distinguish the values $\leq -1, 0, \geq 1$.

In parallel, we pass on the input symbol itself in $\log |\Sigma|$ further dimensions.

Overall, the output $z_t$ of single SSM layer provides, at every position, both the original symbol in $\Sigma$ and an element of $\{\leq -1, 0 \geq 1\}^C$.

We can thus view the output of this layer as a string over an enriched string of symbols $\sigma_1 \times \sigma_2 \in \Sigma \times \{\leq -1, 0 \geq 1\}^C$. Based on this, one can predictively model these languages as follows.

For Dyck-1, the next token is EOS or "(" if $\sigma_2 = 0$, and "(" or ")" after any other prefix (note that predictive modeling assumes valid prefixes).

Shuffle-$k$-Dyck is similar: EOS is allowed if and only if all counters are zero. An opening bracket is always allowed. A closing bracket is only allowed if the respective counter is $> 0$.

For $a^n b^n$, the next token is $a$ or $b$ if $\sigma_1 = a$; $b$ if $\sigma = (a, \geq 1)$ or $(b, \geq 1)$; EOS if $\sigma = (b, 0)$.

Constructions for $a^n b^n c^n$, $a^n b^n c^n d^n$ are similar.

For Boolean expressions, the next token is $\langle n - ARY \rangle$ or EOS if $\sigma_2 = 0$, and any other token otherwise.

All of these constructions can be encoded using an appropriate function $\rho$ applying to $z_t$.     $\square$

## B.5 Bounded-Depth Dyck

**Definition 25.** *The language $Dyck_{K,h}$ [Hewitt et al., 2020, Yao et al., 2021b] is given by the CFG with the nonterminals $\{S_0, S_1, \ldots, S_{h-1}, S_h\}$ and the following production rules:*

$$S_h \rightarrow (_1 S_{h-1})_1 | \ldots | (_K S_{h-1})_K | \varepsilon$$
$$S_{h-1} \rightarrow (_1 S_{h-2})_1 | \ldots | (_K S_{h-2})_K | \varepsilon$$
$$\ldots \ldots$$
$$S_2 \rightarrow (_1 S_1)_1 | \ldots | (_K S_1)_K | \varepsilon$$
$$S_1 \rightarrow (_1 S_0)_1 | \ldots | (_K S_0)_K | \varepsilon$$
$$S_0 \rightarrow \varepsilon$$

*and the start symbol $S_h$.*

**Theorem 26** (Restated from Theorem 6). *There is a two-layer SSM with $d = O(h \log K)$ that predictively models $Dyck_{K,h}$ at all input lengths, at finite precision.*

*Proof.* In the first layer, we calculate each token depth up to $h$ using Lemma 23. After the first layer, at each position, the activations will indicate both the depth up to $h$, and the identity of the symbol. The space of activations is thus $\{0, \ldots, h\} \times \{(_1, )_1, \ldots, (_K, )_K\}$. We then, for each depth $l = 1, \ldots, h$, define a set-reset automaton (Definition 8) given by the set $Q_l := \{l\} \times \{(_1, )_1, \ldots, (_K, )_K\}$. Running all of these set-reset automata will tell us, for each depth, the identity of the last bracket at that depth. We can deduce the maximum depth $h'$ at which the last bracket is an opening one, and thus infer the set of valid next symbols. The activity of these set-reset automata can, in parallel, be simulated by a second SSM layer using Lemma 9. We need $h$ such automata, and each SSM has width $\log K$.  $\square$

# C  Definitions of Languages

Here, we provide formal definitions of languages from the test suite based on Bhattamishra et al. [2020]. Descriptions follow Bhattamishra et al. [2020], and are included here for self-containment. In all cases, our data generation setup is directly taken from [Bhattamishra et al., 2020].

## C.1  Regular Languages

**Tomita Grammars.** Used primarily as a benchmark language family for assessing sequence to sequence models [Tomita, 1982], some of the languages in this family are star-free (with dot-depth of 1) and some non-star-free. All the regular languages of the family are defined on the alphabet $\Sigma = \{0, 1\}$. Individual language definitions are available in Table 1.

**$D_n$.** We follow the definition of Bhattamishra et al. [2020] to define the $D_n$ family of star-free languages. In our experiments, we only generate $D_2$, $D_3$, $D_4$, and $D_{12}$ languages; $D_1$ is equivalent to Tomita-2. All the languages of the family are defined on the alphabet of $\Sigma = \{a, b\}$. $D_n = (aD_{n-1}b)^*$ has level $n$ in the dot-depth hierarchy.

**PARITY.** PARITY is the set of all strings on the alphabet $\Sigma = \{0, 1\}$ such that the number of 1's is even. This language can be easily recognized by a DFA with just two states.

**Others.** We further have the non-star-free languages $(aa)^*$, $(aaaa)^*$ and $(abab)^*$, and the star-free languages $aa^* bb^* cc^* dd^* ee^*$, $\{ab\}^* d\{b, c\}^*$, and $\{0, 1, 2\}^* 02^*$.

## C.2  Counter Languages

**Dyck and Shuffle-Dyck.** Dyck-1 is defined on the alphabet $\Sigma = \{[, ]\}$ and derived using the following CFG production rule: $S \rightarrow (S)|SS|\varepsilon$.

We further use the family of Shuffle-k languages [Suzgun et al., 2019]. Shuffle-Dyck-k is defined in terms of $\Sigma = \{(_1, )_1, \ldots, (_k, )_k\}$. It is defined as the shuffle of $k$ Dyck-1 languages, each defined in terms of the alphabet $\Sigma_i = \{(_i, )_i\}$ where $i = 1, \ldots, k$.

**$n$-ary Boolean Expressions.** This is the set of valid expressions over various operators. We focus on up-to-3-ary expressions, defined using the following grammar:

$S \rightarrow \langle \text{VALUE} \rangle$
$S \rightarrow \langle \text{UNARY OPERATOR} \rangle\ S$
$S \rightarrow \langle \text{BINARY OPERATOR} \rangle\ S\ S$
$S \rightarrow \langle \text{TERNARY OPERATOR} \rangle\ S\ S\ S$

This language is recognized by a counter automaton [Fischer et al., 1968a].

**Others**   We further include the languages of the forms $a^n b^n$, $a^n b^n c^n$, and $a^n b^n c^n d^n$.

| Grammar | Star-Free | Definition |
|---|---|---|
| 1 | Yes | 1* |
| 2 | Yes | (10)* |
| 3 | No | strings without $1^{2n+1}0^{2m+1}$ substrings |
| 4 | Yes | strings without any 000's substrings |
| 5 | No | strings of even length with an even number of 1's |
| 6 | No | strings where number of 0's - number of 1's is divisible by 3 |
| 7 | Yes | 0*1*0*1 |

Table 1: Tomita Grammars

# D   Experimental Details

All experiments used the Mamba reference implementation[8]. xUnless stated otherwise, we followed the defaults given there ( $d_{state} = 16$, $d_{conv} = 4$, $expand = 2$), as we found the default combination to work better than other options. We tuned $d_{model}$ for each language.

## D.1   Test Suite from Bhattamishra et al. [2020]

**Data Preparation**   For all the languages, we use either the data prepared by Bhattamishra et al. [2020] or—where not available—their data-generation scripts, allowing full comparability with results they reported for transformers. We used their official code and data release at `https://github.com/satwik77/Transformer-Formal-Languages` (last commit 48eea2e; MIT license). Training sets typically consist of 10K samples, with lengths varying between 1 to 50. There are two heldout bins: one with in-distribution lengths ([1,50]), and one testing length generalization (lengths [51,100]). The first one was used for hyperparameter optimization. Each bin typically contains around 2K samples. However for languages such as $a^n b^n$, where the number of positive examples in each bin was limited, all possible examples for that bin are included.

**Hyperparameters**   For each language, we conducted extensive hyperparameter search. We varied the $d_{model}$ parameter in Mamba across the set $\{16, 32, 64, 128, 256\}$. Additionally, we experimented with the number of layers in our model, ranging from 1 to 3, training each configuration for 100 epochs. For languages where Mamba performed well, this number of layers was sufficient. However, for languages where Mamba struggled, we increased the number of layers up to 12, with little to no success.

We used the AdamW optimizer. To identify optimal learning rates, we started with a coarse hyperparameter search using values from the set $\{0.001, 0.0001, 0.00001\}$. If one of these learning rates showed high performance, we conducted a more fine-grained search to find the optimal learning rate. Finally, we varied the batch size from $\{16, 32, 64\}$ for datasets with 10K training examples. For languages like $a^n b^n$ with limited training size, we searched for an optimal batch size within the set $\{5, 10\}$.

## D.2   FlipFlop

We obtained the dataset of Liu et al. [2023a] from their release, `https://huggingface.co/datasets/synthseq/flipflop` (MIT license). Our setup corresponds to the deterministic ("clean") mode in Liu et al. [2023a]. Matching Figure 2 in Liu et al. [2023a], we evaluated both with

---

[8]`https://github.com/state-spaces/mamba/blob/main/README.md`

| Language | Model | Bin-1[1, 50] | Bin-2[51, 100] | Bin-3[101, 150] |
|---|---|---|---|---|
| Dyck-1 | Transformer | 100.0 | 100.0 | 100.0 |
| | Mamba1 | 100.0 | 62.6 | 13.91 |
| | Mamba2 | 100.0 | 49.1 | 9.5 |
| | Mamba3 | 100.0 | 53.95 | 10.0 |
| Shuffle-2 | Transformer | 100.0 | 100.0 | 93.0 |
| | Mamba1 | 100.0 | 49.5 | 2.3 |
| | Mamba2 | 100.0 | 61.5 | 8.2 |
| | Mamba3 | 100.0 | 65.5 | 9.7 |
| Shuffle-4 | Transformer | 100.0 | 100.0 | 98.8 |
| | Mamba1 | 100.0 | 44.4 | 4.3 |
| | Mamba2 | 100.0 | 63.8 | 7.2 |
| | Mamba3 | 100.0 | 56.2 | 7.8 |
| Shuffle-6 | Transformer | 100.0 | 99.9 | 94.0 |
| | Mamba1 | 100.0 | 39.4 | 3.4 |
| | Mamba2 | 100.0 | 61.2 | 6.75 |
| | Mamba3 | 100.0 | 59.6 | 9.85 |
| Boolean-3 | Transformer | 100.0 | 100.0 | 99.8 |
| | Mamba1 | 99.75 | 65.7 | 7.05 |
| | Mamba2 | 99.95 | 47.25 | 2.3 |
| | Mamba3 | 100.0 | 73.45 | 8.6 |
| Boolean-5 | Transformer | 100.0 | 99.8 | 99.0 |
| | Mamba1 | 99.9 | 30.05 | 7.6 |
| | Mamba2 | 100.0 | 80.2 | 14.9 |
| | Mamba3 | 99.25 | 60.7 | 6.25 |
| $a^n b^n$ | Transformer | 100.0 | 100.0 | 100.0 |
| | Mamba1 | 100.0 | 4.1 | 0 |
| | Mamba2 | 100.0 | 9.4 | 0 |
| | Mamba3 | 100.0 | 21.3 | 0 |
| $a^n b^n c^n$ | Transformer | 100.0 | 100.0 | 100.0 |
| | Mamba1 | 100.0 | 0 | 0 |
| | Mamba2 | 100.0 | 7.6 | 0 |
| | Mamba3 | 100.0 | 5.1 | 0 |
| $a^n b^n c^n d^n$ | Transformer | 100.0 | 100.0 | 99.4 |
| | Mamba1 | 100.0 | 4.76 | 0 |
| | Mamba2 | 100.0 | 0 | 0 |
| | Mamba3 | 100.0 | 0 | 0 |

Table 2: Accuracies on the counter Languages from the Bhattamishra et al. [2020] test suite. Transformer results reported based on Bhattamishra et al. [2020]. For Mamba, we report best settings (chosen based on inputs of length [1,50]) at 1 (Mamba1), 2 (Mamba2), 3 (Mamba3) layers. Results for the best-performing layer count, from the first two bins, are shown in Figure 5. On these languages, there is also a third bin.

in-distribution data (matching the distribution of the training dataset) with $p_i = 0.8, p_w = 0.1, p_r = 0.1$, and using an out of distribution sparse tail with $p_i = 0.98, p_w = 0.01, p_r = 0.01$, where $p_i, p_w, p_r$ refer to the probabilities of that instruction appearing in input sequences.

We trained a one-layer Mamba with the default parameters[9], setting $d_{model}$ to 16 with the AdamW optimizer using a learning rate of $3x10^{-4}$ and a batch size of 16.

Following the evaluation criteria for LSTMs in Liu et al. [2023a], we compute the test every 100 training steps on our validation sets of choice, by randomly sampling around $10^3$ samples from each set in every evaluation cycle.

---

[9]From https://github.com/state-spaces/mamba/blob/main/README.md

| Language | Model | Bin-1[1, 50] | Bin-2[51, 100] |
|---|---|---|---|
| Tomita 1 | Transformer | 100.0 | 100.0 |
| | Mamba1 | 100.0 | 100.0 |
| | Mamba2 | 100.0 | 100.0 |
| | Mamba3 | 100.0 | 100.0 |
| Tomita 4 | Transformer | 100.0 | 92.4 |
| | Mamba1 | 100.0 | 100.0 |
| | Mamba2 | 100.0 | 100.0 |
| | Mamba3 | 100.0 | 100.0 |
| Tomita 7 | Transformer | 100.0 | 100.0 |
| | Mamba1 | 100.0 | 100.0 |
| | Mamba2 | 100.0 | 100.0 |
| | Mamba3 | 100.0 | 100.0 |
| Tomita 2 | Transformer | 100.0 | 100.0 |
| | Mamba1 | 100.0 | 100.0 |
| | Mamba2 | 100.0 | 100.0 |
| | Mamba3 | 100.0 | 100.0 |
| $aa^*bb^*cc^*dd^*ee^*$ | Transformer | 100.0 | 100.0 |
| | Mamba1 | 100.0 | 100.0 |
| | Mamba2 | 100.0 | 100.0 |
| | Mamba3 | 100.0 | 100.0 |
| $\{a,b\}^*d\{b,c\}^*$ | Transformer | 100.0 | 100.0 |
| | Mamba1 | 100.0 | 100.0 |
| | Mamba2 | 100.0 | 100.0 |
| | Mamba3 | 100.0 | 100.0 |
| $\{0,1,2\}^*02^*$ | Transformer | 100.0 | 68.7 |
| | Mamba1 | 100.0 | 100.0 |
| | Mamba2 | 100.0 | 100.0 |
| | Mamba3 | 100.0 | 100.0 |
| $D_2$ | Transformer | 74.6 | 3.1 |
| | Mamba1 | 100.0 | 100.0 |
| | Mamba2 | 100.0 | 100.0 |
| | Mamba3 | 100.0 | 100.0 |
| $D_3$ | Transformer | 80.9 | 8.5 |
| | Mamba1 | 100.0 | 100.0 |
| | Mamba2 | 100.0 | 100.0 |
| | Mamba3 | 100.0 | 100.0 |
| $D_4$ | Transformer | 90.2 | 3.3 |
| | Mamba1 | 100.0 | 100.0 |
| | Mamba2 | 100.0 | 100.0 |
| | Mamba3 | 100.0 | 100.0 |
| $D_{12}$ | Transformer | 95.18 | 1.5 |
| | Mamba1 | 93.65 | 93.35 |
| | Mamba2 | 99.9 | 95.55 |
| | Mamba3 | 99.99 | 99.85 |

Table 3: Accuracies on the regular Languages from the Bhattamishra et al. [2020] test suite - 1st half. Transformer results reported based on Bhattamishra et al. [2020]. For Mamba, we report best settings (chosen based on inputs of length [1,50]) at 1 (Mamba1), 2 (Mamba2), 3 (Mamba3) layers. Results for the best-performing layer count are also shown in Figure 5.

## D.3 Bounded Hierarchical Structure

We built on the official code and data release of Yao et al. [2021b] at `https://github.com/princeton-nlp/dyck-transformer` (last commit: 5d21fcf). We train a 2-layer Mamba and a 1-layer Mamba on $Dyck_{K,h}$ with $K = 8$ and $h = 10$. The training set and the validation set contains samples of lengths $\leq 700$, while the test set contains samples of lengths $700 \leq n \leq 1400$. We train Mamba with a varying number of layers $l \in \{1, 2\}$ and $d_{model} \in \{20, 30, 40, 50, 60, 70, 80, 90, 100\}$.

| Language | Model | Bin-1[1, 50] | Bin-2[51, 100] |
|---|---|---|---|
| Parity | Transformer | 68.7 | 0 |
| | Mamba1 | 26.95 | 0 |
| | Mamba2 | 80.05 | 4.15 |
| | Mamba3 | 91.15 | 16.7 |
| $(aa)^*$ | Transformer | 100.0 | 0 |
| | Mamba1 | 2.1 | 0 |
| | Mamba2 | 2.1 | 0 |
| | Mamba3 | 4.2 | 0 |
| $(aaaa)^*$ | Transformer | 100.0 | 0 |
| | Mamba1 | 0 | 0 |
| | Mamba2 | 0 | 0 |
| | Mamba3 | 4.0 | 0 |
| $(abab)^*$ | Transformer | 100.0 | 2.5 |
| | Mamba1 | 0 | 0 |
| | Mamba2 | 0 | 0 |
| | Mamba3 | 0 | 0 |
| Tomita 3 | Transformer | 75.4 | 10.8 |
| | Mamba1 | 25.99 | 12.49 |
| | Mamba2 | 36.88 | 17.05 |
| | Mamba3 | 60.85 | 29.37 |
| Tomita 5 | Transformer | 29.3 | 0.0 |
| | Mamba1 | 15.94 | 0 |
| | Mamba2 | 34.5 | 0 |
| | Mamba3 | 38.4 | 0 |
| Tomita 6 | Transformer | 88.8 | 0 |
| | Mamba1 | 7.2 | 0 |
| | Mamba2 | 37.8 | 0 |
| | Mamba3 | 54.56 | 0.04 |

Table 4: Accuracies on the regular Languages from the Bhattamishra et al. [2020] test suite - continued. Transformer results reported based on Bhattamishra et al. [2020]. For Mamba, we report best settings (chosen based on inputs of length [1,50]) at 1 (Mamba1), 2 (Mamba2), 3 (Mamba3) layers. Results for the best-performing layer count are also shown in Figure 5.

We use the Adam optimizer with an initial learning rate of 0.01 or 0.001, using cross-entropy loss. After training for 100 epochs (with early stopping allowed in case of convergence), we select the learning rate with the better training performance.

# E    Finite Precision Assumption

As described in Section 2.1, we adopt the *finite precision* notion used by Weiss et al. [2018]: We allow an unbounded number of integer bits, but only $p$ fractional bits, where $p$ is a sufficiently large constant (e.g., $p = 8$), independent of the length of the input.

There are a variety of related precision notions in the theoretical literature on neural sequence models – here, we discuss the effect of other notions on our results:

1. **Infinite precision** Infinite precision allows any parameter and intermediate value to be an arbitrary number. Such a setting is unrealistic, as it would allow encoding arbitrary detail about the input into infinite precision [e.g. Siegelmann, 1999] and read these out with sufficiently powerful functions ($A$, $B$, $\phi$) in (4) – this would lead to the unrealistic conclusion that any function and language could be represented. For this reason, theoretical work has often adopted restricted precision notions.

2. **Finite inventory of values**, where integer and fractional bits are both restricted. Such a setup may be justified based on the fact that any real computer has bounded memory,

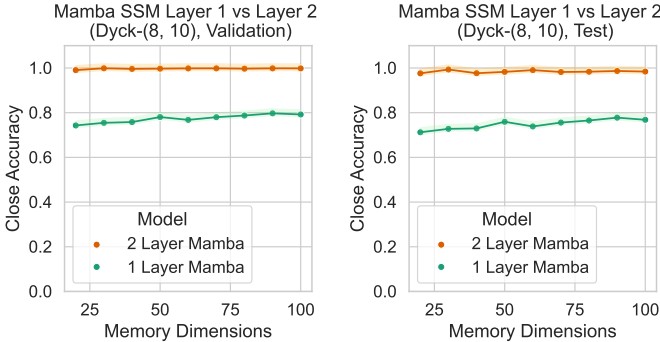

Figure 7: Mamba Accuracy on $Dyck_{8,10}$, on the development set (length $\leq 700$, same length range as training set) and test set (length $700 \leq n \leq 1400$). The latter is also plotted in Figure 4.

though such a setup precludes *any* positive results on non-finite-state problems for *any* computational architecture.[10]

Such a restrictive setup would not affect our positive results on Flip-Flop, Star-Free, and bounded-depth Dyck languages (Theorems 1, 4, 6), as these all use *bounded* finite-precision activation values. As this is a *more* restricted setup than the one we are assuming, this also would not affect our negative results about PARITY and non-star-free languages (Theorems 2, 4). These results are thus highly robust to variations of the finite precision assumption.

Such a more restrictive definition would, however, mean that, for unbounded counting (Theorem 5), modeling is only possible up to a bound determined by the number of possible values—this is the one place where our results would be impacted. Indeed, we do observe that Mamba learns these counter languages on training lengths but struggles with length generalization. Transformers, on the other hand, can represent these languages with bounded activations (due to the constructions in Bhattamishra et al. [2020]), and show strong length generalization.

An intermediary between infinite and finite precision is notions of precision where the number of allowed bits slowly increases with the input length, e.g., logarithmically. Such a setup has particularly been adopted for transformers [Merrill and Sabharwal, 2023], because a finite-precision assumption leads to very low expressivity in transformers. For SSMs, on the other hand, we find that finite precision assumptions are sufficient for showing a broad range of positive results.

---

[10]For instance, a Turing machine with bounded memory and thus a bounded tape is equivalent to a finite-state automaton.

