# OpenReview forum: "The Expressive Capacity of State Space Models: A Formal Language Perspective"
_NeurIPS.cc/2024/Conference — NeurIPS 2024 poster_

### Official Review · Reviewer_px5T · 2024-06-16

**Soundness:** 2
**Presentation:** 3
**Contribution:** 2
**Rating:** 6
**Confidence:** 3

**Summary:**

1. This paper studies the expressive capacity of state-space models from a formal language perspective.
2. It shows that for flip-flop, SSM can do well. And for parity, SSM cannot do well.

**Strengths:**

1. State-space models have the advantage of low inference cost, therefore using it to learn language models and formal languages are important research directions.
2. The perspective of formal language and the proof is interesting

**Weaknesses:**

1. The theorem 2 seems to be problematic:
	1. Given sequence $\mathbf{x} = x_1, \dots, x_T$, consider function $f(\mathbf{x}) = (e^{i \pi \sum_{i=1}^n x_i} + 1)/2$. This continuous function can achieve the goal that if $\sum_{i=1}^n x_i$ is even, then the result is 1, otherwise the result is 0.
	2. Based on a cumulative sum layer, it only requires the approximation using two-layer state-space models for a function $f(x) = (e^{i \pi x} + 1) / 2$ . There are various universal approximation theorem to prove this:
		1. https://openreview.net/forum?id=i0OmcF14Kf
		2. https://arxiv.org/abs/2307.11888
		3. They are also related works in the expressive sense.
		4. Still it is possible to have the unboundedness issue from $\sum_{i=1}^n x_i$.
		5. I understand the notion of finite precision assumption is necessary in practice. However, another question is whether increasing the number of fractional bits p can be a effective method to reduce the problem proposed in this paper. As the cost of increasing p is still linear.  (And increasing p can relax the unboundedness issue in $\sum_{i=1}^n x_i$ in an exponential sense.
		6. I am willing to raise the score if this issue can be better discussed.
	3. The experiments for 28 seem to justify that three layer mamba can work with certain accuracy. A natural question would be whether increasing the hidden dimension and layers improves the performance.
2. The role of parameterization discussed in page 3 seems a bit lacking of related work
	1. HIPPO(https://arxiv.org/abs/2008.07669), Approximate Diagonalization(https://arxiv.org/abs/2310.01698), StableSSM(https://arxiv.org/abs/2311.14495).
3. Line 682, typo - have shown.

**Questions:**

1. I understand the notion of finite precision assumption is necessary in practice. However, a natural question is whether increasing the number of fractional bits p can be a effective method to reduce the problem proposed in this paper. As the cost of increasing p is still linear.
2. In the same argument, it seems transformer and temporal convolution, in the setting of convolution, also have the same problems as Theorem 2. I tend to believe this issue comes from the setup of finite precision and infinite sequence length. Any insight about this issue?

**Limitations:**

1. The result derived in this paper is limited to formal languages. It is not clear what is the generalization to natural languages.

---

> ### Comment · Reviewer_px5T · 2024-08-03
> **Related work, just a comment**
>
> Given the formal nature of the paper’s title, it may be beneficial to reference the related paper available at https://arxiv.org/abs/1606.06737, which discusses formal language and the hidden Markov model. As the state-space model is a specific type of hidden Markov model, a brief discussion on the relationship between star-free state tracking, probabilistic regular grammar, and context-free grammar would be a valuable addition to the paper.

---

> > ### Author Response · Authors · 2024-08-07
> >
> > Thanks. We will be happy to cite this paper, and briefly discuss how star-free state tracking relates to probabilistic regular and context-free grammars. Our results on Dyck languages are particularly relevant in relation to the latter.

---

> ### Author Rebuttal · Authors · 2024-08-06
>
> We appreciate the reviewer's detailed feedback and constructive criticism. We would like to clarify/address the issues raised as follows:
>
> ### Weakness
> **(1.1), (1.2)** We thank the reviewer for the interesting construction for PARITY, and the universal approximation references.
>
>   We would like to clarify why the suggested construction isn't a contradiction to Theorem 2. One could implement the suggested construction in an SSM-like architecture.  However, computing the function $f(x) = (e^{i \pi x}+1)/2$ requires a layer-wise nonlinear function (eg an MLP) implementing sine/cosine functions at arbitrarily large inputs. Importantly, to obtain a construction working for all input lengths (as our positive results do), a single such operation would need to work at arbitrarily large input numbers.
>
> As stated in line 56, our analysis assumes the layer-wise operations to be linear or (Swi)GLU (line 56) (as these are used in the real-world SSMs we surveyed in Appendix A). This is relevant to the proof of Theorem 2. A single GLU or SwiGLU is likely not able to represent sine/cosine for unboundedly large inputs, and we believe the same holds for more classical MLPs (ReLU/sigmoid MLPs). This is because, to the extent we are aware, universal approximation results for feedforward networks guarantee approximation in the sense of compact convergence [1], uniformly on the compactification of $\mathbb{R}$, [2], or in $L^p$ [3], none of which cover uniform approximation of sine/cosine on $\mathbb{R}$.  Indeed, very recently, [4] studied universal approximation in $C_b(\mathbb{R})$, exactly what would be needed for approximating sine/cosine at any input magnitude. We believe their Proposition 5.5 entails that sine/cosine cannot be uniformly approximated with certain activation functions.
>
> Thus it's not to be expected that $f(x) = (e^{i \pi x}+1)/2$ would be implemented by a typical MLP uniformly for arbitrarily large inputs. Hence, we believe any implementation of the proposed construction would either require custom (e.g., periodic) activation functions or require model size to vary with the input length, both resolving an apparent contradiction with Theorem 2.
>
> Thanks further for linking the references on universal approximation, which we will cite and discuss in the Related Work section.
>
> We would like to clarify again, there's no contradiction of our negative result in Theorem 2 to these universal approximation results. Both Wang et al and Orvieto et al provide universal approximation guarantees, but importantly, the size of the approximating networks depends on input length. One can see this from the statements of the core formal results, Propositions 3.6 and 3.9, in Wang et al. It is clearly stated by Orvieto et al in their Remark 2. In contrast, our results (both positive and negative) concern the existence of a *single* SSM recognizing a formal language at *any input length*.
>
>   Overall, we thank the reviewer again for raising these interesting points. We will address them as follows:
>
>   (1) Cite and discuss Wang et al and Orvieto et al results.
>
>   (2) Make more salient, & mention in the Theorem 2 statement that it relies on properties of the activation functions as stated in line 56.
>
>   (3) Discuss the question whether SSMs could benefit from more general, e.g., periodic activation functions in the nonlinearities.
>
>  We thank the reviewer for this insightful observation and for helping improve our paper with such constructive criticism.
>
> **(1.3)** For languages where Mamba failed to predictively model perfectly with 3 layers, we did increase number of layers to 12 and hidden dimension to 256, as documented in Appendix Section D.1. Across the languages tested, we obtained no further benefits beyond 3 layers.
>
> **(2)** In the "Role of Parameterization" section, we aimed to delineate the scope of our work and point out that our results aren't primarily affected by technical details regarding parameterization, as we allowed each of A, B, and $\rho$ to be arbitrary functions. However, we acknowledge the importance of citing relevant papers that focus on parameterization and will make our point about our results being unaffected by parameterization details more explicit.
>
> **(3)** We thank the reviewer for pointing out the typo. We'll fix it.
>
> ### Questions
>
> **(1)** We agree that the question of precision is interesting. Increasing $p$ to any finite bound might improve expressiveness on short inputs, but not mitigate limitations on unboundedly long inputs. An open question is to what extent increasing $p$ with input length ($N$), e.g., $\log N$ could mitigate expressiveness limitations in theory. We will add discussion of the same.
>
> **(2)** Indeed, both transformers and convolutional networks face similar challenges as SSMs regarding PARITY. For transformers, [5] shows that such limitations *provably* persist even with infinite precision (PARITY creates very sharp minima on long inputs). We conjecture that this reflects a general phenomenon in highly parallellizable architectures, and that similar phenomena may apply with SSMs even with infinite precision.
>
> ### Limitations
> We acknowledge that our results are derived for formal languages. However, Dyck, Flip-Flop, $a^nb^n$ etc. have linguistic motivations and provide insights into the structure/processing of natural language, which we'll expand on in the camera ready version.
>
> ### References
> [1] Cybenko, G. (1989). Approximation by superpositions of a sigmoidal function. Mathematics of control, signals and systems
>
> [2] Ito, Y. (1992). Approximation of continuous functions on Rd by linear combinations of shifted rotations of a sigmoid function with and without scaling. Neural Networks
>
> [3] Arora, et al (2018) "Understanding Deep Neural Networks with Rectified Linear Units." ICLR
>
> [4] van Nuland, T. D. (2024). Noncompact uniform universal approximation. Neural Networks
>
> [5] Hahn and Rofin (2024). Why are Sensitive Functions Hard for Transformers? ACL

---

> > ### Comment · Reviewer_px5T · 2024-08-12
> >
> > Thank you for the detailed explanation. My issue has been thoroughly and clearly resolved. I’ve updated my score from 4 to 6.

---

> > > ### Author Response · Authors · 2024-08-12
> > >
> > > We are pleased to hear that your concerns have been addressed. Thank you so much for your feedback and suggestions

---

### Official Review · Reviewer_9mjL · 2024-06-18

**Soundness:** 3
**Presentation:** 2
**Contribution:** 4
**Rating:** 7
**Confidence:** 4

**Summary:**

The paper presents several results on the expressiveness of *state space models (SSMs)*, viewed as *language acceptors* or a weak form of *language predictors*, from the perspective of formal language theory. These results are parametrized to subsume currently common SSM architectures, including *non-negative SSMs*, which encode prefixes using positive reals only, *time-invariant SSMs*, where prefixes are encoded independent of the currently considered token, and *finite precision SSMs*, meaning that internal computations are limited by a fixed number of fractional bits.

In more detail, the paper presents the following results:
- The FlipFlop language is predicted by a two-layer SSM with finite precision (Theorem 1), and there is no non-negative, finite precision SSM that recognizes the PARITY language (Theorem 2).
- The class of non-negative, finite precision SSMs predicts exactly the star-free languages (Theorem 4).
- Various kinds of languages that require unbounded counting, such as Dyck languages, $a^nb^nc^n$, or boolean expressions, are predicted by an SSM.

In summary, the paper establishes foundational results categorizing the expressive power of SSMs within the well-established framework of formal language theory. Additionally, this allows for a comparison of SSM with their imminent competitor, transformers, for which similar architectures exist. This comparison is empirically supported by experiments.

**Strengths:**

The paper considers the expressiveness of recent SSMs from the perspective of formal language theory. While it is not new to analyze sequence-classifying or sequence-to-sequence models from a formal language viewpoint, this paper is one of the first to do so for SSMs on a broad scale. Additionally, this is a reasonable continuation of preceding work, particularly for understanding the differences in expressiveness between SSMs and transformers.

The main result, at least from my perspective, Theorem 4, states that SSMs predict exactly the star-free languages. This is fundamental, making it well worth considering. The remaining results are more specialized, focusing on particular languages. However, these also provide insights for a more detailed understanding of the expressive power of SSMs, especially compared to transformer. Thus, these are of clear relevance as well.

In general, all results are supported by technical proofs and some intuition building, allowing the reader to verify the soundness of the results in sufficient detail without unreasonably extensive effort.

**Weaknesses:**

The clarity and quality of the paper could be improved. This is mainly due to three reasons:
- This first reason is minor compared to the others. There are many results in this paper, which is desirable. However, it is a bit difficult to get an overview of the results. For example, it is unclear which should be considered the main result. Of course, it could be the case that there is not a single main result, which I assume applies here. However, the structure of the results seems odd, as we first show what is (not) possible for some languages (Theorem 1, Theorem 2), then consider a general result (Theorem 4), and then, again, focus on specific languages (Theorem 5 and Theorem 6). I think the main problem is that all results are squashed into a single “Theoretical Results” section, which is also mixed with basic definitions like finite-state automata and (star-free) regular languages. Giving the reader a better hint on the central theme of the results and a clear separation of preliminaries and results would improve the clarity of the paper.

- The authors tend to state lemmas and theorems informally, making it hard to grasp the concrete setting. For example, Theorem 2 states, *”… can recognize PARITY at arbitrary input lengths …”*. It is not precisely clear what this means, especially the “arbitrary length” part. Or take Lemma 13 in Appendix B.3. This lemma states that *”… the character $w_{t-1}$ can be read out from $z_t$ at finite precision.”*. It is hard to understand what is exactly meant by this, too. The same goes for the overall statement of Lemma 22.
While I appreciate that the main part of the paper focuses on a low-technical presentation, such a theoretical work cannot be separated from its technical appendix. Therefore, statements in the appendix should be precise. This goes beyond these three examples and is a general problem. This also applies to the exact SSM architecture considered in respective results. In most cases, one can guess what is meant, but I don’t think such things should be left to the reader’s imagination to this degree.

- Some proofs could be more detailed. For example, the proof of Lemma 17 includes previously unused notation (line 824, $w_{1..t} = …00$), it is not clear how equalities (15) and (16) are derived or what *”… running these in parallel, …”* means.

**Questions:**

- The notion of finite precision you are using seems to have been considered before. Could you elaborate on why it is a reasonable one?
At first sight, it seems artificial, as you either consider practical settings where you are limited by some amount of bits for representing a number, or you do not care for practical realizations and thus everything can be unbounded. The notion you used feels a bit like a convenience choice, as fixing fractional parts helps keeping fuzziness under control but still allows for unbounded counting using unbounded integer parts. Especially, in the context of your positive, predictive results regarding finite precision. I assume these do not hold for an overall bounded number of bits.

**Limitations:**

The authors state the limitations of their work, mainly focusing on the lack of experimental evaluation, which is sufficient regarding this part. However, I would have appreciated a clearer statement on open questions focusing on the theoretical contributions. For example, they could discuss related work on other models like RNNs or Transformers and which results about formal language recognition or prediction exist in those contexts, which have not yet been considered in the context of SSMs.

---

> ### Author Rebuttal · Authors · 2024-08-06
>
> We thank the reviewer for their positive score and encouraging review. Regarding the issues pointed out:
>
> ### Response to Weaknesses:
>
> > This first reason is minor compared to the others. There are many results in this paper, which is desirable. However, it is a bit difficult to get an overview of the results. For example, it is unclear which should be considered the main result. Of course, it could be the case that there is not a single main result, which I assume applies here. However, the structure of the results seems odd, as we first show what is (not) possible for some languages (Theorem 1, Theorem 2), then consider a general result (Theorem 4), and then, again, focus on specific languages (Theorem 5 and Theorem 6). I think the main problem is that all results are squashed into a single “Theoretical Results” section, which is also mixed with basic definitions like finite-state automata and (star-free) regular languages. Giving the reader a better hint on the central theme of the results and a clear separation of preliminaries and results would improve the clarity of the paper.
>
> - The paper indeed does not have a single main result; rather, we have four major results that we aimed to make explicit and contextualize in the Take-Aways section on Page 9. We acknowledge that combining some of the background information with our main results in the Theoretical Results section was not ideal. We will separate the background information from the results into distinct sections to improve clarity.
>
>
> > The authors tend to state lemmas and theorems informally, making it hard to grasp the concrete setting. For example, Theorem 2 states, ”… can recognize PARITY at arbitrary input lengths …”. It is not precisely clear what this means, especially the “arbitrary length” part. Or take Lemma 13 in Appendix B.3. This lemma states that ”… the character can be read out from at finite precision.”. It is hard to understand what is exactly meant by this, too. The same goes for the overall statement of Lemma 22. While I appreciate that the main part of the paper focuses on a low-technical presentation, such a theoretical work cannot be separated from its technical appendix. Therefore, statements in the appendix should be precise. This goes beyond these three examples and is a general problem. This also applies to the exact SSM architecture considered in respective results. In most cases, one can guess what is meant, but I don’t think such things should be left to the reader’s imagination to this degree.
>
>  > Some proofs could be more detailed. For example, the proof of Lemma 17 includes previously unused notation (line 824, ), it is not clear how equalities (15) and (16) are derived or what ”… running these in parallel, …” means.
>
> - We agree that understanding the concrete settings of our results might require the reader to refer to the appendix. We appreciate that the reviewer acknowledges our focus on intuition in the main paper, leaving technical details in the appendix. We also thank the reviewer for pointing out that some of our proofs, although correct, might require more detail. For each theorem or lemma, we will provide a fully formally precise statement in the Appendix. We will revisit all proofs and provide additional steps to better elucidate them.
>
>
>
>
> ### Response to Questions:
>
> > The notion of finite precision you are using seems to have been considered before. Could you elaborate on why it is a reasonable one? At first sight, it seems artificial, as you either consider practical settings where you are limited by some amount of bits for representing a number, or you do not care for practical realizations and thus everything can be unbounded. The notion you used feels a bit like a convenience choice, as fixing fractional parts helps keeping fuzziness under control but still allows for unbounded counting using unbounded integer parts. Especially, in the context of your positive, predictive results regarding finite precision. I assume these do not hold for an overall bounded number of bits.
>
>
> We understand that our choice of finite precision for representing fractional components versus allowing unbounded counting, and thus unbounded integer values, might seem like a convenience choice. However, we would like to clarify that we do not consider unbounded integer values in any of our theorems except for the unbounded counting case: All our results except for the unbounded counting case in fact have overall bounded numbers of bits. For unbounded counting, the task definition requires unbounded integer values, and any length-generalizing recurrent solution will require an unbounded overall number of bits. While, theoretically, a solution for unbounded counting can be constructed with SSMs, as shown in our paper, it might be unlikely that such a solution would be practical owing to such requirements of unbounded overall number of bits. Indeed, in our experiments across all counter languages in Figure 3, we were unable to get the SSM to learn a length-generalizable solution. We will expand our discussion of the precision assumptions and their implications in Appendix E, and add appropriate notes in the main paper.

---

> ### Comment · Reviewer_9mjL · 2024-08-08
>
> Thank you for your clarifications. I appreciate your willingness to improve the clarity of some of your proofs.
>
> However, I have a follow-up question as I am not fully satisfied with your comment on your definition of "finite precision":
>
> If I understand your comment correctly, you imply that results like Theorem 4, which are not the "unbounded counting case" are also valid if we assume "full" finite precision, meaning an overall bounded number of bits for representing numerical values?

---

> > ### Author Response · Authors · 2024-08-08
> >
> > Yes, that's correct. Our positive results on Flip-Flop, Star-Free, and bounded-depth Dyck languages (Theorems 1, 4, 6) would not be affected and remain valid even under the more restricted definition of finite precision, i.e., "full" finite precision. Similarly, our negative results regarding Parity and non-star-free languages would also remain unaffected (Theorems 2, 4). Only Theorem 5 (unbounded counting) would be invalid under full finite precision. We discuss this briefly in Appendix E (lines 1046 to 1068). We hope this clarifies your question. However, please let us know if further details are needed on this issue.

---

> > > ### Comment · Reviewer_9mjL · 2024-08-13
> > >
> > > Thanks for getting back to me.
> > >
> > > I stand with my initial rating and recommend this submission to accepted.

---

> > > > ### Author Response · Authors · 2024-08-13
> > > >
> > > > Thank you so much for the discussion, and your suggestions. We are grateful for your recommendation

---

### Official Review · Reviewer_tByQ · 2024-07-09

**Soundness:** 3
**Presentation:** 4
**Contribution:** 4
**Rating:** 8
**Confidence:** 5

**Summary:**

This paper presents a comprehensive theoretical and empirical analysis of the expressive capacity of modern State Space Models (SSMs) within the framework of formal languages and automata theory. The authors establish important theoretical results, demonstrating that SSMs can effectively model star-free languages, various counter languages, and bounded hierarchical structures without relying on explicit stack mechanisms. By leveraging the Krohn-Rhodes theorem and concepts from algebraic theory, the paper provides rigorous proofs showing the capabilities and limitations of SSMs in capturing different language classes. By connecting theoretical computer science (TCS) with machine learning (ML), the paper highlights the importance of understanding the fundamental limitations and capabilities of computational models.

**Strengths:**

- The paper introduces a novel theoretical framework that connects formal language theory with the expressive capabilities of State Space Models (SSMs), a relatively unexplored area in machine learning research.
- By leveraging the Krohn-Rhodes theorem and algebraic theory, the authors provide proofs and characterizations of the language classes that SSMs can model, including star-free languages and bounded hierarchical structures.
- The integration of TCS concepts with modern SSM architectures is important because through TCS we study the limits and what is possible or not possible
- Theoretical contributions are rigorously presented, with clear and detailed proofs supporting the claims about the expressive power of SSMs. I checked all the proofs thoroughly and I am sure they are valid.

**Weaknesses:**

- While the paper does an admirable job of explaining complex theoretical concepts, some sections may still be challenging for readers without a strong background in theorem proving. Additional intuitive explanations or visual aids could help make the content more accessible.
- In the related work section add more works that study this, in particular: Limits of Deep Learning: Sequence Modeling through the Lens of Complexity Theory. Nikola Zubić, Federico Soldá, Aurelio Sulser, Davide Scaramuzza - https://arxiv.org/abs/2405.16674 . Mention also how does your work overlap with it and is there any interesting connection?
- There are typos, use for example Grammarly to fix them. Also there are some typos in math, for example Lemma 22 should have w_j inside the sum, not w_i etc. Check and make sure indexes are well defined. Theorems are proven well, but there are typos.

**Questions:**

- While the experiments on synthetic datasets and formal languages are comprehensive, additional experiments on more complex and diverse real-world tasks could strengthen the empirical validation. Are there any plans to extend the experimental validation to other types of datasets or applications that are not necessarily big, but challenging and prove the points?
- The paper briefly mentions the practical implications of the theoretical insights for designing new ML architectures. Could the authors provide concrete examples or guidelines on how to implement these insights in real-world systems? What are the key considerations for practitioners looking to apply these findings?

**Limitations:**

Already addressed by authors.

---

> ### Author Rebuttal · Authors · 2024-08-06
>
> We thank the reviewer for their high score and positive comments about our paper. We appreciate the reviewer's thorough examination of our proofs and their constructive insights and suggestions. We address the points raised as follows:
>
> ### Response to Weaknesses:
>
> > While the paper does an admirable job of explaining complex theoretical concepts, some sections may still be challenging for readers without a strong background in theorem proving. Additional intuitive explanations or visual aids could help make the content more accessible.
> -  We acknowledge that some sections of the paper may be challenging for readers without a strong background in theorem proving. To make the content more accessible, we will include additional intuitive explanations and visual aids. Specifically, we will add visual representations to help elucidate the SSM constructions described in our theorems.
>
> > In the related work section add more works that study this, in particular: Limits of Deep Learning: Sequence Modeling through the Lens of Complexity Theory. Nikola Zubić, Federico Soldá, Aurelio Sulser, Davide Scaramuzza - https://arxiv.org/abs/2405.16674 . Mention also how does your work overlap with it and is there any interesting connection?
>
> - The paper by Zubić et al. indeed represents significant work in the field of analyzing expressivity of SSMs. Their approach differs from ours in that they focus on a range of algorithmic tasks, such as function composition, whereas we focus on fine-grained study of length-generalizable expressions for core formal languages. We do not see any overlap in results, but will make sure to include the comparison in the related work section of our main paper.
>
>
> > There are typos, use for example Grammarly to fix them. Also there are some typos in math, for example Lemma 22 should have w_j inside the sum, not w_i etc. Check and make sure indexes are well defined. Theorems are proven well, but there are typos.
>
> We apologize for the typographical errors and mistakes in the mathematical notation, such as the incorrect index in Lemma 22. We will thoroughly proofread the paper  again and carefully correct all the typos we missed.
>
>
> ### Response to Questions:
> > While the experiments on synthetic datasets and formal languages are comprehensive, additional experiments on more complex and diverse real-world tasks could strengthen the empirical validation. Are there any plans to extend the experimental validation to other types of datasets or applications that are not necessarily big, but challenging and prove the points?
> - We agree that additional experiments on more complex and diverse real-world tasks would further strengthen our empirical validation, and we do plan to do the same in follow up work.
>
> > The paper briefly mentions the practical implications of the theoretical insights for designing new ML architectures. Could the authors provide concrete examples or guidelines on how to implement these insights in real-world systems? What are the key considerations for practitioners looking to apply these findings?
> - One of the main practical implications that can be taken from the theoretical insights of our paper in designing new ML architectures is for future architectures to combine the strengths of attention with that of the SSM recurrence update, and indeed some work [1][2][3] that has been published since our submission has found empirical evidence of the same. We do agree that more concrete examples and guidelines could be given and would include some examples in the appendix of our paper due to space considerations.
>     - [1] Lieber, Opher, et al. "Jamba: A hybrid transformer-mamba language model." arXiv preprint arXiv:2403.19887 (2024).
>     - [2] Waleffe, Roger, et al. "An Empirical Study of Mamba-based Language Models." arXiv preprint arXiv:2406.07887 (2024).
>     - [3] Ren, Liliang, et al. "Samba: Simple Hybrid State Space Models for Efficient Unlimited Context Language Modeling." arXiv preprint arXiv:2406.07522 (2024).
>
> We once again appreciate the reviewer’s positive comments and valuable feedback and will incorporate the suggestions to enhance the clarity, completeness, and practical relevance of our paper.

---

> ### Comment · Reviewer_tByQ · 2024-08-08
>
> 1. Authors said that they "will include additional intuitive explanations and visual aids" in the paper.
> 2. Zubić et al. work will be included in the Related Work section.
> 3. Authors said that they "will thoroughly proofread the paper again and carefully correct all the typos we missed".
> 4. In the follow-up work they can also do more experiments for various tasks.
> 5. Authors discussed that probably Attention + SSMs will be the future of Neural Net Architectures.
>
> Therefore, everything I had as problem is addressed, and I will finalize my rating as:
>
> 8: Strong Accept

---

> > ### Author Response · Authors · 2024-08-08
> >
> > We are pleased to hear that your concerns have been addressed and are grateful for your strong acceptance rating. Thank you so much for your feedback and suggestions.

---

### Official Review · Reviewer_6tRo · 2024-07-11

**Soundness:** 3
**Presentation:** 2
**Contribution:** 3
**Rating:** 7
**Confidence:** 2

**Summary:**

This study investigates the expressive power of SSMs compared to Transformers and RNNs from the perspective of formal language classes (or circuit complexities), and reports their distinct strengths. Previous studies have shown that both SSM and Transformers are in TC0, suggesting some state tracking problems are hard for both models. This study shows that both models cover overlapping but distinct parts of TC0. Theorem 1 claims that two-layer SSM can predictively model the Flip Flop (a star-free language) state tracking, and Theorem 2 claims that no SSM with non-negative gates can recognize PARITY (a non-star-free language). Since Flip-Flop and PARITY are building blocks of star-free and non-star-free languages, Theorems 1 and 2 are summarized to Theorem 4 claiming that SSMs with non-negative gates can predictively model a regular language L iff it is star-free. Further Theorems 5 and 6 claim several context-free and context-sensitive are predictively modeled by SSMs. Experimental results support theoretical claims.

Due to the limitation of resources, I did not check the proofs. In particular, I am not familiar with formal language theory and I did not have enough time to learn about it. This is a __suggestion for chairs__ for future avoidance of mismatches, that the reviewers should be __examined__ if they have fundamental knowledge/background/understandings in the field. I am an expert of expressive power analysis, but not at all of formal language theory. Unfortunately, this kind of mismatch happens every year, so I am usually skeptical to any mathematical ''theorems'' published in machine learning conferences.

**Strengths:**

- Previous studies are precisely reviewed.
- Theorem 4 strictly refines the previous results that SSMs are in TC0, a subclass of regular languages.
- Theorem 5 and 6 are new results in context-free and context-sensitive languages.

**Weaknesses:**

I am really interested in reading the manuscript. Just because I am not familiar with formal language theory, some expressions were a bit hard to parse for me. For example,
- in Eq.4, what does the subscript of w_{1…t} mean? In the previous sentence, w does not have any subscript. Are they the same or distinguished?
- at l.215: “L is non-star-free if and only if recognizing it involves counting modulo some K.” What is K?
- It would improve the readability if the authors could explain what is TC0. In my opinion, NeurIPS audiences (are familiar with sample complexity but) are less familiar with computational complexity.
- What is Chomsky–Schützenberger theorem?

**Questions:**

- Is the assumption of non-negative gates critical in Theorem 4? If so, can Mamba be improved by replacing non-negative gates (such as ReLU or Swish) with signed gates (such as Leaky ReLU)?

- Some authors claim that neural networks are Turing complete. For example,

  [PMB19] _Jorge Pérez, Javier Marinković, Pablo Barceló, On the Turing Completeness of Modern Neural Network Architectures, ICLR2019_

  If I understand correctly, TC0 is a subclass of regular language, and strictly smaller than Turing machine. What are essential differences from their claims?

---

> ### Author Rebuttal · Authors · 2024-08-06
>
> We thank the reviewer for their thoughtful comments and questions. We address the raised points as follows:
>
> ### Response to Weaknesses:
>
> - Equation 4 Notation: We acknowledge that the subscript $w_{1...t}$ in Equation 4 was intended to indicate that the prefix $w$ consists of a sequence of $t$ inputs, but it is superfluous. We will revise the notation for consistency and clarity, matching the representation in the previous sentence.
> - Modulo Some K: The reference to "modulo some K" at line 215 indicates that $K$ is a natural number ($K \in \mathbf{N}$). For instance, in the case of Parity, $K$ is 2 because we use modulo 2 to determine the sequence output.
> - Explanation of TC0: While TC0 is a common term in computational complexity literature, we agree that it might not be familiar to everyone in the diverse NeurIPS audience. Although space constraints prevented us from including a detailed explanation in the main paper, we will add a brief explanation in the appendix to improve readability.
> - Chomsky–Schützenberger Theorem: The Chomsky–Schützenberger theorem is a foundational result in formal language theory. It states that any context-free language can be represented using a combination of a Dyck language and a regular language. More formally, any context-free language can be expressed as a homomorphic image of the intersection of a Dyck language with a regular language.
>
> ### Response to Questions:
>
> > Is the assumption of non-negative gates critical in Theorem 4? If so, can Mamba be improved by replacing non-negative gates (such as ReLU or Swish) with signed gates (such as Leaky ReLU)?
>
> - Our proof for Theorem 4 does make the assumption of non-negative gates. With more general gates, we show in Theorem 21 (page 22 in the Appendix) that expressiveness would increase substantially.  In Mamba, the non-negativity is created by parameterizing gates as exponentials. We have attempted to change the implementation to allow signed gates, but found training to stop working, suggesting the non-negativity may be useful for training. We believe that studying SSMs without the nonnegativity constraint presents an interesting oppportunity for follow-up work.
>
> > Some authors claim that neural networks are Turing complete. For example, [...] If I understand correctly, TC0 is a subclass of regular language, and strictly smaller than Turing machine. What are essential differences from their claims?
>
> - Regarding the Pérez et al. paper, it is indeed a seminal piece of work, and we do cite it in our paper. Its results show the Turing completeness of transformers when they are allowed to generate an unbounded number of intermediate tokens before making their decision -- akin to chain-of-thought and scratchpad techniques. This is a key difference to our setup: We are interested in the expressiveness of neural models when they have to make their decision immediately upon processing the input, without additional intermediate steps. This is foundational because any additional steps would result in higher computational cost. Indeed, there is work [1][2] showing that transformers become more expressive than TC0 when allowed intermediate steps (while increasing computational cost due to intermediate steps) -- and we believe that the same arguments will hold for SSMs.
>
> [1] Li et al, Chain of Thought Empowers Transformers to Solve Inherently Serial Problems, ICLR 2024
>
> [2] Feng et al, Towards Revealing the Mystery behind Chain of Thought: A Theoretical Perspective, NeurIPS 2023

---

> > ### Comment · Reviewer_6tRo · 2024-08-12
> >
> > Thank you for your detailed response. I would like to increase my score based on your responses to my questions. I am convinced that the theoretical contributions are worthy enough to be published now.
> >
> > It would be very helpful for readers to add Responses to Questions to the paper/appendix.

---

> > > ### Author Response · Authors · 2024-08-12
> > >
> > > We will include our responses to the questions raised in the appendix for clarity, as suggested. We are delighted to know that we were able to convey the importance of our theoretical results and are grateful to the reviewer for increasing their score.

---

### Author Rebuttal · Authors · 2024-08-06

We appreciate the reviewers' detailed feedback and constructive criticism. We are particularly encouraged by Reviewer tByQ's thorough verification of our proofs and confirmation of their validity. We also recognize the valuable input from all reviewers regarding typographical errors and structural improvements to enhance the clarity of the paper. Below, we summarize the common points raised and outline our planned revisions.

We acknowledge the need for improved readability and structure in the results section. To address this, we will separate the background information from the theoretical results section, providing a clearer overview as suggested by Reviewer 9mjL. This restructuring will help highlight the four major results explicitly, as outlined in the "Takeaways" section on Page 9. Additionally, we will revise the notation in Equation 4 to ensure consistency throughout the paper.

Several reviewers pointed out the need for clearer explanations of specific concepts. We will add brief explanations of terms like TC0 and the Chomsky-Schützenberger theorem to the appendix to enhance readability for the diverse NeurIPS audience, as suggested by Reviewer 6tR0. Furthermore, in response to Reviewer tByQ's suggestion, we will include visual aids to better elucidate our proofs, complementing the theoretical details with intuitive images to help convey the core ideas of the paper.

We are particularly grateful to Reviewer px5T for suggesting a construction for PARITY. As we described in our response, this construction would not be length-generalizable -- thus resolving an apparent contradiction with Theorem 2. We will include a more detailed discussion in our camera-ready version, and thank Reviewer px5T for this insightful contribution.

We also appreciate the suggestions from Reviewers px5T and tByQ to cite additional relevant papers. We will incorporate these references into our related work section, providing a more comprehensive view of the ongoing research on the expressivity of State Space models.

Regarding concerns about our assumption of the finite precision setting, we have addressed each individual query in our responses. We are happy to clarify further if there are any additional questions on this topic.

In summary, we are grateful for the reviewers' valuable insights, which have significantly contributed to improving our work. We believe these revisions will enhance the quality and clarity of our paper, and we look forward to presenting the final version.

---

### Decision · Program_Chairs · 2024-09-25

**Decision:**

Accept (poster)

**Comment:**

Paper provides a novel and an intriguing theoretical analysis of the expressive capacity of SSMs, a timely and relevant topic given the recent success of SSMs in language modeling. The authors meticulously connect SSM capabilities to formal language theory, offering valuable insights into the strengths and limitations of these models.

While this paper's rigorous theoretical analysis of SSM through the lenses of formal language theory and its intriguing insights into the model's capabilities are highly valuable, the following weaknesses should be addressed to broaden its impact and reach a larger audience within the community: improving the clarity and accessibility of certain sections (reviwers initially found some sections challenging to parse), particularly for readers without a strong background in formal language theory, and expanding the empirical validation beyond synthetic datasets to include more complex and diverse real-world tasks.

The authors have committed to improving the acessibility by adding intuitive explanations, visual aids, and clarifying technical details in the appendix, and we expect these commitments to be fulfilled.